# Lysosomal integral membrane protein-2 as a phospholipid receptor revealed by biophysical and cellular studies

Karen S. Conrad [1], Ting-Wen Cheng[2], Daniel Ysselstein[3], Saskia Heybrock[4], Lise R. Hoth[1], Boris A. Chrunyk[1], Christopher W. am Ende[1], Dimitri Krainc[3], Michael Schwake[3], Paul Saftig[4], Shenping Liu[1], Xiayang Qiu[1] & Michael D. Ehlers[2,5]

Lysosomal integral membrane protein-2 (LIMP-2/SCARB2) contributes to endosomal and lysosomal function. LIMP-2 deficiency is associated with neurological abnormalities and kidney failure and, as an acid glucocerebrosidase receptor, impacts Gaucher and Parkinson's diseases. Here we report a crystal structure of a LIMP-2 luminal domain dimer with bound cholesterol and phosphatidylcholine. Binding of these lipids alters LIMP-2 from functioning as a glucocerebrosidase-binding monomer toward a dimeric state that preferentially binds anionic phosphatidylserine over neutral phosphatidylcholine. In cellular uptake experiments, LIMP-2 facilitates transport of phospholipids into murine fibroblasts, with a strong substrate preference for phosphatidylserine. Taken together, these biophysical and cellular studies define the structural basis and functional importance of a form of LIMP-2 for lipid trafficking. We propose a model whereby switching between monomeric and dimeric forms allows LIMP-2 to engage distinct binding partners, a mechanism that may be shared by SR-BI and CD36, scavenger receptor proteins highly homologous to LIMP-2.

[1] Medicinal Sciences, Pfizer Worldwide R&D, Eastern Point Road, Groton, CT 06340, USA. [2] Neuroscience Research Unit, Pfizer Worldwide R&D, 610 Main Street, Cambridge, MA 02139, USA. [3] Department of Neurology, Northwestern University Feinberg School of Medicine, Chicago, IL 60611, USA. [4] Biochemical Institute, Christian-Albrechts University Kiel, Olshausenstrasse 40, D-24098 Kiel, Germany. [5] Present address: Biogen, 225 Binney St., Cambridge, MA 02142, USA. Karen S. Conrad and Ting-Wen Cheng contributed equally to this work. Correspondence and requests for materials should be addressed to S.L. (email: Shenping.liu@pfizer.com) or to X.Q. (email: Xiayang.Qiu@pfizer.com)

LIMP-2 is a type III transmembrane glycoprotein encoded by the *SCARB2* gene primarily found in lysosomes and late endosomes[1]. Mutations in *SCARB2* have been identified as causative for action myoclonus-renal failure syndrome (AMRF)[2–4], a form of progressive myoclonic epilepsy associated with severe renal dysfunction[5]. AMRF can present with renal and/or neurological features with 14 disease-causing *SCARB2* mutations identified to date[6].

LIMP-2 is a receptor for the trafficking of glucocerebrosidase (GCase) from the endoplasmic reticulum to lysosomes, where the acidic environment activates the sphingolipid processing capability of GCase[7]. Deficiency in GCase activity causes Gaucher disease[8] and is implicated in Parkinson's disease[9, 10]. LIMP-2 loss-of-function mutations result in reduced lysosomal GCase activity with consequent accumulation of the GCase substrate glucosylceramide, glucosylsphingosine and the fibril-forming protein α-synuclein, characteristic pathological markers of Gaucher disease and Parkinson's disease, respectively[11]. In addition to its role as a GCase receptor, LIMP-2 was identified as a receptor for enterovirus 71 and coxsackievirus A16, which cause hand, foot, and mouth disease in humans; an illness which can be associated with severe neurological symptoms[12].

LIMP-2 belongs to the class B scavenger receptor family that also includes CD36 and SR-B1. CD36 and SR-B1 are primarily receptors for fatty acids (FAs)[13] and high-density lipoproteins (HDL)[14], respectively, but their known ligands have expanded to include collagen[15], low-density lipoproteins (LDL)[16], and phospholipids[17]. Importantly, these proteins can facilitate selective, endocytosis-independent uptake of lipids[14, 18]. Moreover, like LIMP-2, these scavenger receptors interact with pathogenic and parasitic protein partners: CD36 binds the malaria parasite PfEMP1 protein[19], and SR-B1 the hepatitis C virus capsid[20]. Two lines of evidence suggest that, like CD36 and SR-B1, LIMP-2 may also specifically function in lipid trafficking: (1) overexpressing LIMP-2 results in cholesterol (CLR) containing enlarged hybrid endosome/lysosome compartments[21]; (2) LIMP-2 knockout (KO) mice have a cellular phenotype in the inner ear and ureter reminiscent of an impaired membrane trafficking and tubular proteinemia[22].

All reported crystal structures of LIMP-2[23–25] and CD36[26], to date, are monomers; however, dimeric and oligomeric complexes of these two proteins have been detected in cells[23, 27]. An α-helical bundle in these proteins consisting of helices α4, α5, and α7 is critical for interacting with protein partners including GCase and enterovirus 71 capsid[7, 23–25, 28–30]. A hydrophobic tunnel, reminiscent of one in the cholesteryl ester transfer protein[31] traverses the length of both proteins and is thought to end near the plasma membrane. This tunnel in CD36 was suggested to be involved in the trafficking of insect pheromone[32]. In LIMP-2, lipid tunnel access was proposed to be mediated by pH-induced movement observed in the critical α-helical bundle[24, 25]. A lipid uptake mechanism involving this tunnel will require large conformational changes since the tunnel is largely blocked in these monomeric structures based on our analysis.

Here, we investigate the potential lipid trafficking function of LIMP-2. By determining a crystal structure of a lipid-bound dimeric form of LIMP-2 luminal domain, and by testing LIMP-2 binding to liposomes using surface plasmon resonance (SPR), negative stain electron microscopy (NS-EM), and dynamic light scattering (DLS), we show that LIMP-2 binds phospholipids much like SR-BI and CD36[17]. Furthermore, we demonstrate in cellular assays that LIMP-2 is a receptor capable of mediating phospholipid uptake. Taken together, these results establish LIMP-2 as a lipid receptor.

## Results

**Structure determination of a lipid-bound LIMP-2 dimer.** We determined a lipid-bound structure of the LIMP-2 luminal domain expressed in HEK cells at 3.0 Å resolution (Table 1; Fig. 1 and Supplementary Fig. 1). Two important differences exist between this structure, previous LIMP-2 structures[23–25], and the structure of FA-bound CD36[26]. First, LIMP-2 in our structure forms a symmetric dimer (root-mean-square deviation 0.98 Å for all Cα atoms in the dimer after applying twofold symmetry, Fig. 1a) via contacts in its two highly distinctive regions of the protein, the α-helical bundle and the region that forms the hydrophobic tunnel (Fig. 1b). Second, clear electron densities for one phosphatidylcholine (PC) molecule and one CLR molecule were present at the hydrophobic tunnel within each monomer (Fig. 1c). CLR is completely, and PC partially, buried inside the hydrophobic tunnel (Fig. 1b). These lipids were presumably acquired from the expression system. The CLR molecules have well defined electron densities (Supplementary Fig. 1B). While the buried head groups and the *SN2* acyl chains of PC molecules have well defined electron densities, the *SN1* acyl chains are outside of the hydrophobic tunnel and are modeled without defined electron densities (Supplementary Fig. 1C). The orientations of the PC molecules at the hydrophobic tunnel are further supported by the specific interactions described below between their head groups and the hydrophobic tunnel. Elongated electron densities were also observed at the hydrophobic clefts described below; one allowed modeling of a fragment of a putative PC molecule (Supplementary Fig. 1D). Several peptides in the LIMP-2 dimer become disordered due to conformational changes caused by lipid binding and dimerization (Fig. 1a).

The LIMP-2 dimer has the shape of a parallelepiped prism of dimensions ~73 × 63 × 40 Å with two clefts on the sides and a

**Table 1 X-ray crystallography data collection and refinement statistics**

| Data collection | |
|---|---|
| Resolution (Å) | 98.0–3.0(4.24–3.0)[a] |
| Space group | $P4_22_12$ |
| Unit cell constants | 139.04 Å 139.04 Å 178.28 Å 90° 90° 90° |
| $R_{merge}$ | 0.151(0.631) |
| $R_{meas}$ | 0.164(0.686) |
| $R_{pim}$ | 0.063(0.266) |
| $I/\sigma(I)$ | 13.7(4.0) |
| $CC_{1/2}$ | 0.999(0.981) |
| Completeness (%) | 99.9(99.9) |
| Redundancy | 12.4(12.3) |
| Refinement | |
| Resolution (Å) | 40.61–3.00 |
| No. reflections | 34139 |
| $R_{work}$ / $R_{free}$ | 0.195/0.230 |
| No. atoms | |
| Protein | 6475 |
| Ligand/ion (specify/describe) | 426 glycans, 169 lipids |
| Water | 0 |
| B factors | |
| Protein | 131 |
| Ligand/ion | 163 |
| Water | N/A |
| R.m.s. deviations | |
| Bond lengths (Å) | 0.010 |
| Bond angles (°) | 1.27 |

One crystal was used for data collection. PDB code: 5UPH
[a] Values in parentheses are for highest-resolution shell

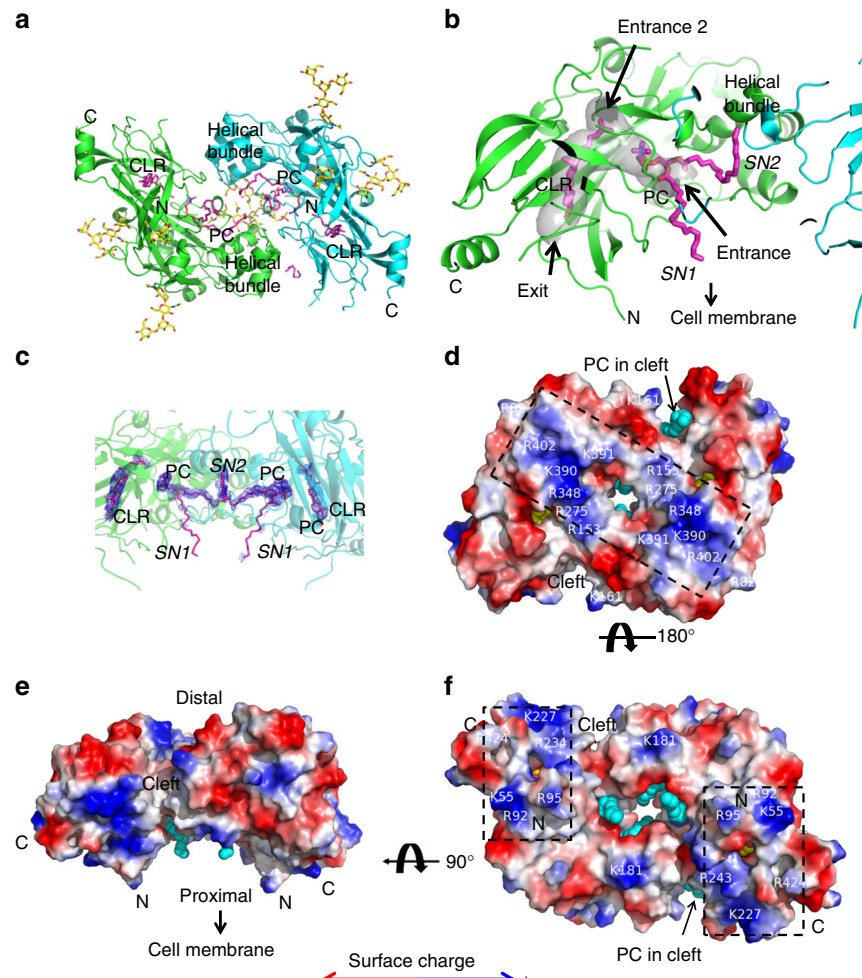

**Fig. 1** Structure of lipids bound LIMP-2 dimer. **a** Lipid-bound LIMP-2 adopts a dimeric structure. The twofold axis of the dimer is perpendicular to the plane of the figure at the center. Protein ribbons are colored green and cyan for different subunits. Endogenous ligands, recognized as phosphatidylcholine (PC) and cholesterol (CLR), are shown in stick model, with carbon atoms colored magenta, oxygen red, nitrogen blue, and phosphorus orange. Carbon atoms in glycan moieties are colored yellow. Image prepared using pymol[60]. The N and C termini and the helical bundles are labeled. **b** Zoomed-in view of one of the hydrophobic tunnels (calculated using program caver[61]) in the LIMP-2 dimer; the entrances/exit to the hydrophobic tunnel are indicated by arrows. Bound lipids with their relative orientation to the cell membrane, and the helical bundle of the second subunit are indicated. Exit and entrance 2 are the proximal and distal openings highlighted in Fig. 2d. **c** PC and CLR molecules in the hydrophobic tunnel are shown embedded in 2mFo-DFc composite omit map (blue mesh) contoured at 1.0σ. **d**–**f** Surface electrostatic potential presentations of the LIMP-2 dimer. Color bar indicates blue positive and red negative charges, with neutral shown in white. Bound ligands in view are shown in yellow spheres for CLR, and cyan for PC, and cleft bound PC is indicated. Surface cationic patches are enclosed in dashed boxes and residues forming these patches are labeled. **d** The putative distal face of the LIMP-2 dimer. **e** Side view of the dimer. Orientation of the luminal domain relative to the cell membrane is indicated. **f** The putative cell membrane proximal face of the LIMP-2 dimer. Notice that CLR molecules are visible through openings on both the distal (**d**) and the proximal (**f**) faces. **a** and **f**, and **c** and **e** have the same views, respectively. For clarity, glycans are omitted in **d**–**f**

central cavity (Fig. 1d–f; Supplementary Movie 1). Based on the locations of the N-termini and C-termini, which connect the transmembrane segments in the full-length LIMP-2, the opposing faces of the prism can be described as the distal face (Fig. 1d) and membrane proximal face (Fig. 1f), respectively. Surface electrostatic potential analysis indicated that both faces have cationic patches next to hydrophobic areas (Fig. 1d–f), a combination that favors interaction with phospholipid bilayers. These cationic patches are formed by clusters of basic residues, R82, R153, R275, R348, K390–K391, and R402 on the distal surface, and K55, K92, R95, K181, K227, K234, and R424 on the proximal face (Fig. 1d, f). These clusters of basic residues are on the surface of the LIMP-2 monomer as well, but dimerization combines them into one larger entity and forms the additional clefts between the two subunits.

To further identify the endogenous phospholipids found in the structure, we analyzed our LIMP-2 samples using liquid chromatography–mass spectrometry (LC/MS) and liquid chromatography–tandem mass spectrometry (LC–MS/MS). From the analysis of the spectra, varieties of PC and small contents of phosphatidylethanolamine (PE) molecules were detected (Supplementary Table 1; Supplementary Fig. 2). Dissolved phosphatidylserine (PS) used as LC/MS standard was detectable, but not found in the LIMP-2 samples used for generating crystals, suggesting it was not available to the secreted LIMP-2 molecules. CLR did not ionize well under the conditions of this analysis, and its presence was neither confirmed nor contradicted.

In agreement with previous work[23], the LIMP-2 luminal domain in solution exists mainly as a mixture of monomer, dimer and some higher order oligomers based on size exclusion

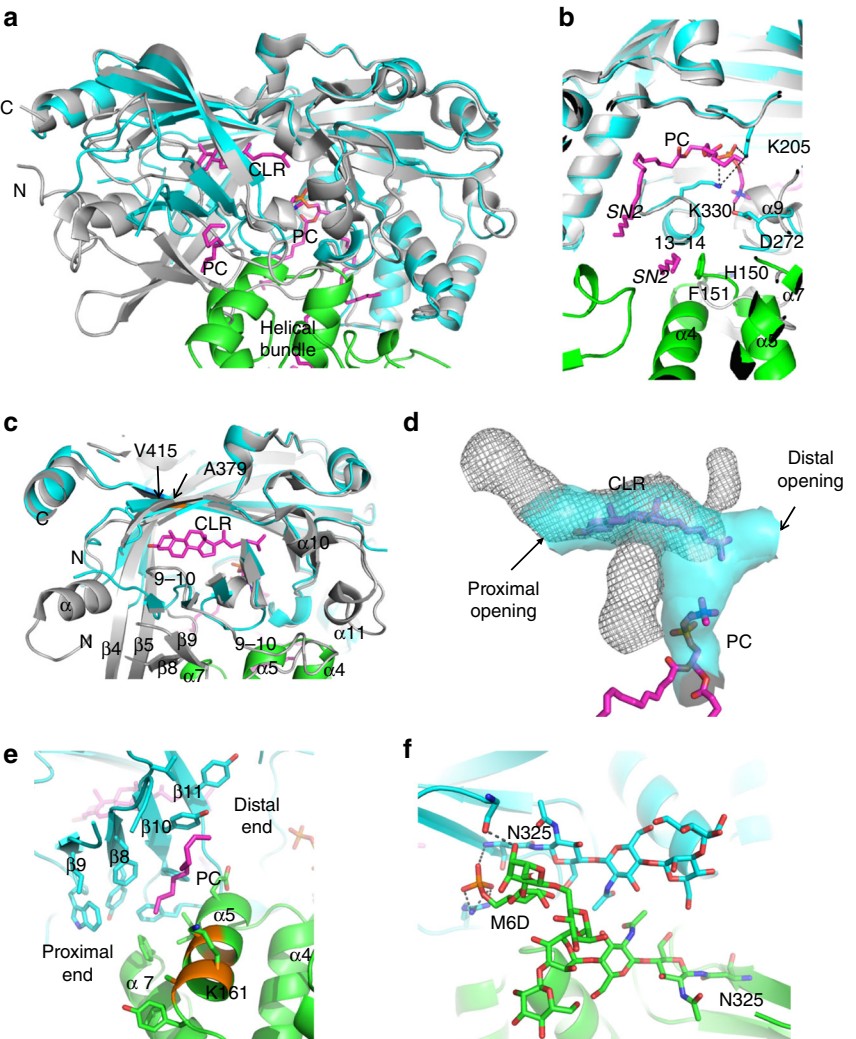

**Fig. 2** Ligand binding sites and dimerization interface of LIMP-2 compared to apo LIMP-2 monomer structure. **a** Each subunit (cyan ribbons) of the LIMP-2 dimer has extensive conformational changes compared to those of the apo LIMP-2 monomer (gray ribbons, PDB code 4Q4F). The helical bundle of the second subunit (α4, α5, and α7 in green ribbons) is shown to indicate the location of dimer interface. **b** The PC-binding site at the hydrophobic tunnel. The head group of bound PC displaces the α9 helix from its position in the apo structure, and interacts with K205, D272, and K330 (labeled). Hydrogen bonds are shown as dashes. Residues chosen for dimerization interfering mutations (H150 and F151) are indicated. **c** Extensive conformational changes observed surrounding the CLR-binding site. Residues V415 and A379, the equivalent residue proposed to be in the cholesterol path in SR-B1, are highlighted. Regions that are affected by CLR binding are labeled. **d** The hydrophobic tunnel of the apo LIMP-2 (gray mesh) is significantly different in shape and location from that of the LIMP-2 dimer (cyan solid surface) upon CLR and PC binding. **e** Hydrophobic cleft and the putative bound PC fragment, with contacting residues side chains shown in sticks. Equivalent residues that are important for HDL uptake by SR-BI or fatty acid or oxLDL uptake by CD36 are colored orange on the main chain. **f** The expansive mannose-6 glycans (M6D), P-Man9GlcNAc2 at N325, contribute to dimer interactions. Hydrogen bonds involving N-glycan are shown in dashes

chromatography (SEC) (Supplementary Fig. 3A), native and SDS gels (Supplementary Fig. 3B), and multi-angle light scattering (MALS) experiments (Supplementary Fig. 3C). Both the monomer and the dimer species are demonstrably stable after initial separation on SEC, and do not interconvert in solution after prolonged storage (Supplementary Fig. 3B, C). The lack of re-equilibration of these species is most likely due to the facts that lipid binding strongly favors dimer and bound lipids may not diffuse easily into aqueous solution, while the monomer is the energetically stable form of the apo protein. Indeed, phospholipids were detected in all dimer samples tested even after extensive detergent wash and hydrophobic interaction chromatography, while they were significantly less abundant in monomer samples (Supplementary Fig. 3D), and their relative contents in the monomer roughly track with the contents of residual dimer detected by native gels (Supplementary Fig. 3B).

**Ligand binding sites and dimerization interface.** Compared to the reported apo structures[23–25], each LIMP-2 subunit has extensive conformational changes that accommodate lipid binding and facilitate protein dimerization (Fig. 2a and Supplementary Movie 2). The PC-binding site is at the entrance of the hydrophobic tunnel (Fig. 1), with the head group pointed inward (Fig. 2b). Near the PC head group, D272 and helix α-9 (residues 270–274) are flipped out of the channel to anchor and accommodate the quaternary amine of PC, while K330 and K205 are shifted to contact the phosphodiester of PC (Fig. 2b). The flipped α9 interacts with the α-helical bundle of the opposite subunit. The *SN1* acyl chains of PC exit the hydrophobic tunnel entrance toward the proximal direction and lack specific interactions with LIMP-2 (Fig. 1b). The *SN2* chains bend back to the distal direction toward the dimer interface (Fig. 1b) and have hydrophobic interactions with the protein and with each other (Fig. 2b).

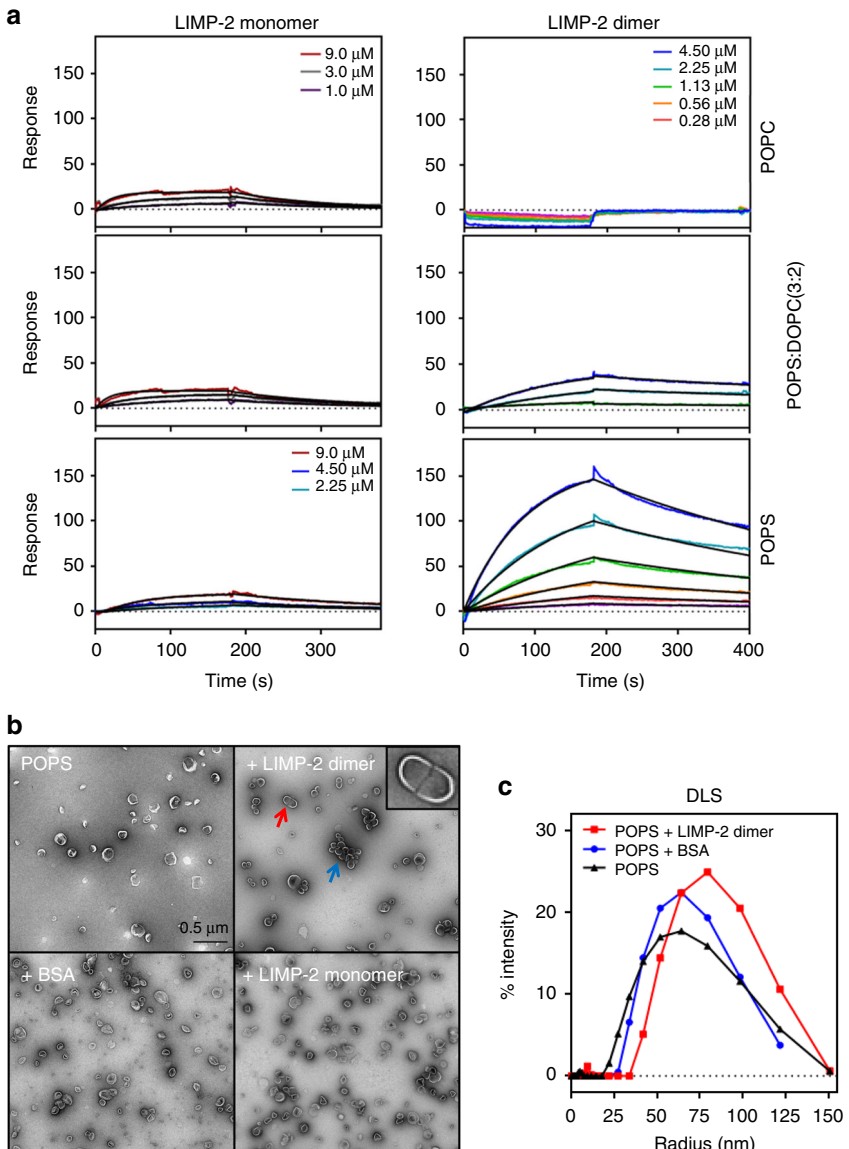

**Fig. 3** LIMP-2 binding to liposomes characterized by SPR, NS-EM, and DLS. **a** Binding kinetics of LIMP-2 to liposomes at pH 7.5 were characterized by SPR. Representative SPR sensorgrams (colored traces) of LIMP-2 monomer (left) and dimer (right) binding to immobilized POPC (top panel), DOPS:DOPC (3:2) (middle) and POPS liposomes (bottom) at concentrations indicated were overlaid with kinetic fitting curves (black) using 1:1 kinetic model. The binding responses are scaled according to dimer-POPS responses. Immobilized levels of POPC, DOPS:DOPC (3:2), and POPS on the L1 SPR chip were 8900, 7500, and 6000 units, respectively. **b** Examples of negative stain EM micrographs indicate the LIMP-2 dimer bridges POPS liposomes and induces clustering of liposomes. Red arrow indicates a clear example of two liposomes forming a tight association with a clear boundary bridged by LIMP-2 dimer (insert is enlarged view). Blue arrow indicates a grape bundle shaped cluster of multiple liposomes. These features are not observed in any images of BSA and LIMP-2 monomer samples, which were used as negative controls. Total numbers of images observed were 5, 8, 16, and 12 for BSA + POPS, monomer + POPS, POPS, and POPS + LIMP-2 dimer, from duplicated experiments. **c** Addition of LIMP-2 dimer into POPS liposome (red) shifts the DLS profiles of POPS liposome (black). POPS liposome with BSA (blue) used as a negative control. The average radii are $50.6 \pm 0.4$, $6.53 \pm 0.06$, $68.6 \pm 1.1$, $3.84 \pm 0.02$, $52.4 \pm 4$ Å for PS liposome, LIMP-2 dimer, PS + LIMP-2 dimer, BSA, and PS + BSA, respectively from 15 measurements of each duplicated experiments. The estimated molecular weights of POPS and POPS + LIMP-2 dimer are $32,774 \pm 1300$ and $66,557 \pm 2500$ KD, respectively

The CLR molecule is located in the lower half of the hydrophobic tunnel (Fig. 1b). To accommodate CLR, the long loop connecting β9 and β10 (9–10, residues 232–265) flips away and is entirely displaced by the α-helical bundle of the opposite subunit (Fig. 2a, c). Additional peptides are disordered or displaced due to CLR binding, including α1 (47–55), β4 (100–106)–β5 (107–118), β8 (212–216)–β9 (226–231), and α10 (298–302) through α11 (308–312) (Fig. 2c). In SR-BI, mutagenesis and covalent inhibitor labeling (residues G420 and C384 in SR-B1, corresponding to V415 and A379 in

LIMP-2) indicated that the equivalent CLR site is critical for the selective uptake of CLR[23, 33–36]. CLR makes extensive hydrophobic interactions with the hydrophobic tunnel of LIMP-2, and there are no polar interactions involving its hydroxyl group (Fig. 2c).

The shape of the hydrophobic tunnel changes significantly upon CLR and PC binding (Fig. 2d). The displacements/disorder of 9–10, α1, α10, and α11 (Fig. 2c) make the CLR site a much less restricted channel with large openings that connects the distal and the proximal faces of the dimer (Figs. 1d, f and 2d). The

entrance to the tunnel also shifts (Fig. 2d) due to the shift of α9 (Fig. 2b).

The cleft of the LIMP-2 dimer at which the additional PC molecule locates is formed by clusters of hydrophobic residues from α5 and α7 of one subunit, and β8–11 of the opposing subunit (Fig. 2e). At the edge of this site is a basic residue, K161, which potentially interacts with phospholipid polar head groups. When a covalent inhibitor labeled the equivalent of K161 in CD36 (K164), the FA uptake of CD36 was inhibited[37]. This site along α5 is known to be involved in HDL binding for SR-B1 and oxidized LDL binding for CD-36[23], and both proteins are known to facilitate selective uptake of phospholipids[18]. In the monomer, this site is exposed on the surface of these proteins; in the dimer, the hydrophobic residues on β8–11 are exposed after the conformational changes (Fig. 2a) and form the hydrophobic clefts with α5 and α7 (Figs. 1d and 2e). These hydrophobic clefts can serve as channels that connect two faces of LIMP-2 to potentially facilitate uptake of lipids, suggesting an active dimer mechanism that is common for these homologous proteins.

Dimerization covers ~900 Å$^2$ protein surface of each LIMP-2 subunit. The main protein–protein contacts at the dimer interface are between the C-termini of α4 and α7, the N-terminus of α5, and the associated loops of one subunit, and loop 10–11 (α9 in the apo structure) and loop 13–14 (328–336) (Fig. 2b) of the opposing subunit. Conformational changes of α9 also contribute to formation of this protein–protein interface (Fig. 2b) and binding of PC at the hydrophobic tunnel (Fig. 2d). In addition to these protein–protein interactions, the acyl chains of the PC molecules bound at both the hydrophobic tunnel (Fig. 2b) and the clefts (Fig. 2e) also contribute extensively to the stabilization of LIMP-2 dimer, explaining the requirement of lipid binding to support dimerization. Re-arrangements of β8–9 are needed for the second subunit to approach the first subunit (Fig. 2c) and to form the PC-binding hydrophobic clefts (Fig. 2e). The expansive mannose-6 glycan, P-Man9GlcNAc2 at N325, makes intermolecular contacts in the dimer (Fig. 2f). The SEC profile of LIMP-2 expressed in glycosylation deficient cells (HEK293-GnTi-) shifted significantly toward elution at monomeric molecular weight compared to that of the protein expressed in wild-type (WT) cells (HEK293F) (Supplementary Fig. 3A). Furthermore, dimerization-deficient mutants, H150A and H150A/F151D, designed based on structure analysis (Fig. 2b) also demonstrated large decreases in the molecular weights (3.0 and 9.5 kDa, respectively) measured by SEC–MALS in comparison to WT, significant larger than the experiment uncertainties (<0.8 kDa) (Supplementary Fig. 4A). The amount of associated lipids also decreased significantly in dimerization-deficient mutant proteins (Supplementary Fig. 4B).

In the structure of FA-bound CD36[26], the conformation of CD36 monomer is similar to that of the LIMP-2 monomer, but deviates significantly from those of the subunits of LIMP-2 dimer (Supplementary Fig. 5A). Although two bound FA molecules in CD36 occupy similar positions in the hydrophobic tunnel as the CLR and the PC in LIMP-2 dimer (Supplementary Fig. 5B–D), openings of the CLR/FA site at both the distal and proximal faces are blocked (Supplementary Fig. 5B, C). FAs do not have the head group of PC to shift the α9 loop and provide less bulk than the CLR does to shift the 9–10 loop (Supplementary Fig. 5D–E). In addition, the FA species in CD36 do not have the long *SN2* acyl tails of PC that directly contribute to the dimerization of LIMP-2. The hydrophobic tunnel of lipid-bound LIMP-2 is much shorter and less restricted than that of FA-bound CD36 (Supplementary Fig. 5F). For CD36 and SR-BI to form and function as dimers, CLR, and phospholipids are likely needed to occupy their hydrophobic tunnels to induce similar conformational changes observed in LIMP-2.

**LIMP-2 dimer binds PS liposomes but not GCase.** We used SPR to study the binding of LIMP-2 in both monomeric and dimeric forms to immobilized liposomes as a means to investigate LIMP-2/lipid interactions in cells[38], and to GCase. Monomer and dimer forms used in SPR were separated initially from size exchange chromatography, further purified by ion exchange chromatography, and were characterized using native and SDS–PAGE gels, SEC–MALS and lipids content analysis (Supplementary Fig. 3). We separately evaluated monomeric and dimeric LIMP-2 binding to liposomes consisting of (i) 1-palmitoyl-2-oleoyl-*sn*-glycero-3-phosphocholine (POPC); (ii) 1-palmitoyl-2-oleoylphosphatidyl-serine (POPS), and (iii) 1,2-dioleoyl-*sn*-glycero-3-phospho-L-serine:1,2-dioleoyl-*sn*-glycero-3-phosphocholine (3:2) (DOPS: DOPC).

At pH 7.5, the LIMP-2 monomer and dimer behaved very differently in binding to immobilized liposomes (Fig. 3a). LIMP-2 dimer-bound exclusively to anionic PS liposomes with robust responses that follow a 1:1 binding kinetic model (Fig. 3a and Supplementary Table 2). LIMP-2 monomer, on the other hand, bound to different liposomes much more weakly (Fig. 3a), and the weaker binding responses (<1 per 10–15 of those dimer at the same concentrations at the end of association phases, Fig. 3a) cannot be explained by its smaller molecular weight (1 per 2.3 of the dimer, Supplementary Fig. 3C). Analysis based on the different binding responses and the contents of the residual dimer in the monomer (Supplementary Fig. 3D) indicated that binding responses of putative monomer to PS liposomes was primarily due to residual dimeric species in the samples used, and can be fitted with contributions from only the dimer, resulting in very similar binding kinetics but a fraction of $R_{max}$ value. Furthermore, the binding of monomer LIMP-2 to PC liposomes, already weak at high protein concentrations and high liposome levels, was undetectable when protein concentrations and liposome levels were decreased (Supplementary Fig. 6). The dependence on both high PC liposome immobilization levels and protein concentrations for observations of weak binding responses of LIMP-2 monomer are typical for binding that requires oligomerization. Determination of these weak binding affinities of the LIMP-2 monomer to liposomes was not meaningful under conditions tested.

At pH 5.0, LIMP-2 monomer and dimer both had greater binding responses for immobilized PS-containing liposomes than at pH 7.5, with the dimer-PS binding demonstrating the highest responses (Supplementary Fig. 7). This is also consistent with SEC–MALS experiments in which there are marked shifts in monomer/dimer equilibrium toward dimer/multimer at pH 5.0 (Supplementary Fig. 3C). These results indicate that in the acidic lysosomes LIMP-2 should have higher binding affinities toward PS liposomes than at a neutral pH on the cell membrane.

When liposomes were placed in the mobile phase for SPR-binding measurements, they exhibited strong binding to immobilized LIMP-2 at both pH 7.5 and 5.0 after subtraction of background binding. These observations could be attributed to avidity effects resulting from the large sizes of the liposomes, and were effectively irreversible with nearly undetectable rates of dissociation (Supplementary Fig. 8). In a situation that LIMP-2 could pack in the cell membrane as densely as on the SPR chip, which is highly unlikely (see below), both LIMP-2 monomer and dimer would bind PC and PS liposomes through avidity effects of multiple LIMP-2 molecules binding to liposomes simultaneously.

We additionally tested LIMP-2 binding to PS liposomes using NS-EM and DLS. LIMP-2 dimer, but not monomer, strongly induced clustering of PS liposomes (Fig. 3b), indicating that LIMP-2 luminal domain dimer can tether lipid bilayers. This observation was quantified by DLS measurements that

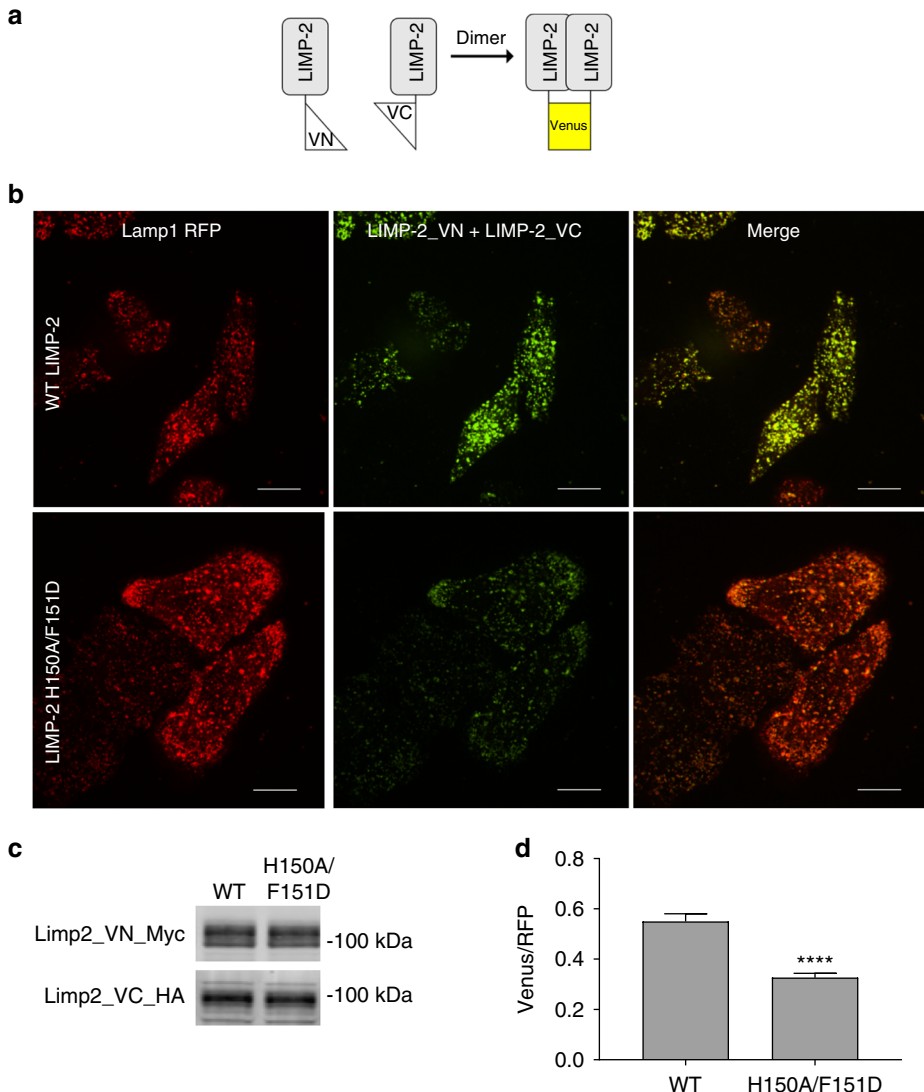

**Fig. 4** Detection of LIMP-2 dimer in live cells using bimolecular fluorescence complementation method. **a** Schematic illustration of constructs designed for bimolecular fluorescence complementation in examining dimerization of LIMP-2. **b** High-resolution confocal images of Hela cells expressing LIMP-2 (WT, top, or H150A/F151D, bottom) fused to the N-terminal and C-terminal fragments of Venus fluorescent protein along with Lamp1-RFP co-transfected as a lysosome colocalization marker. **c** Western blot showing expression levels of the LIMP-2 WT and H150A/F151D variants using anti-myc and anti-HA antibodies. **d** Quantification of the ratio of mean Venus fluorescence intensity relative to ER-localized mCherry for cells transfected with LIMP-2 (WT and H150A/F151D) fused to the N-terminal and C-terminal fragments of Venus fluorescent protein fluorescence. Data is expressed as mean ± SEM, ~60 cells each experiment, $n = 3$, ****$P < 0.0001$, $t$ test. Scale bar 20 µm

demonstrated an increase in particle sizes in PS liposome suspensions when LIMP-2 dimer was added (Fig. 3c).

LIMP-2 monomer and dimer had opposite behaviors when binding to immobilized GCase, as measured by SPR. LIMP-2 monomer bound GCase strongly, as reported[7, 24], but LIMP-2 dimer had a minimal binding response, which can be attributed to the minor monomer component in the sample (Supplementary Fig. 9A). The conclusion that only LIMP-2 monomer binds to GCase is further supported by SEC–MALS experiments in which the LIMP-2 monomer was shown to form a 1:1 complex with GCase, but LIMP-2 dimer did not (Supplementary Fig. 9B).

The binding behaviors of LIMP-2 monomer and dimer support the observed LIMP-2 dimer structure. The cationic patches on both the distal and the proximal faces of the LIMP-2 dimer may simultaneously bind, effectively tethering, two anionic PS liposomes. The LIMP-2 monomer also has these two cationic patches, but the dimer binds much stronger due to avidity effects of combining two monomers. Alternative to this structural

explanation, we cannot rule out that the monomer may only have one lipid interacting site, but the dimer tethers PS liposomes by combining two sites on the opposite faces of the dimer. The monomer has an additional weak binding site for PC liposomes, likely at the entrance of the hydrophobic tunnel to facilitate entry of PC molecules into the hydrophobic tunnel. This PC site is blocked in the dimer by the bound PC molecules. Surface cationic patches may interact with PS liposomes, but may not be well suited for binding individual lipids, which may provide another explanation why PS is not detected in our samples. In the LIMP-2 dimer, the α-helical bundle which is known to be critical for binding to GCase[7, 23, 24, 30] is close to the dimer interface and thereby unavailable for GCase binding.

**Detection of LIMP-2 dimer in cells.** Heterologously expressed full-length LIMP-2 was reported to be able to dimerize/oligo-merize with LIMP-2 luminal domain[23]. We also detected

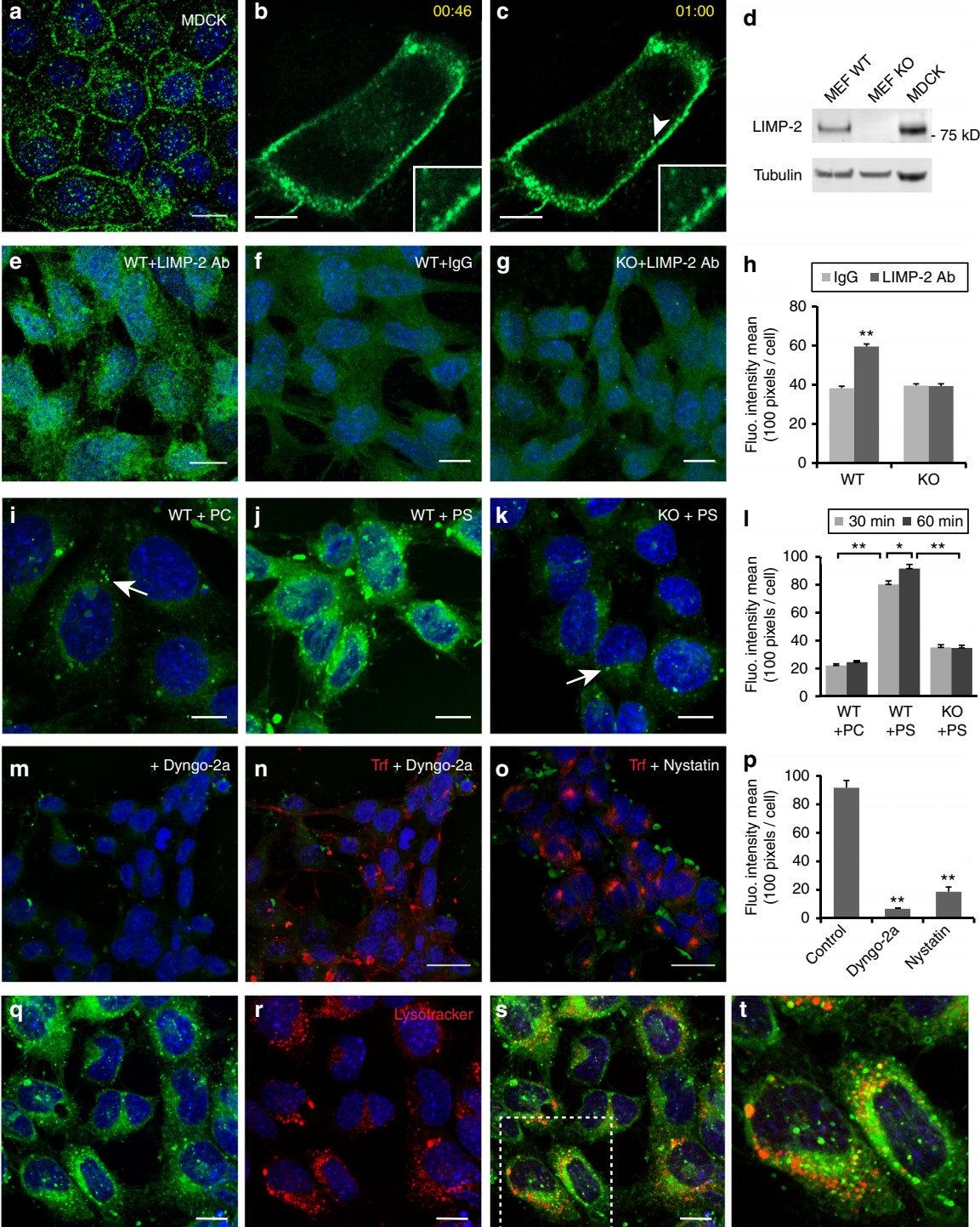

**Fig. 5** LIMP-2 mediates the endocytosis of liposomes in cells. **a** Immunocytochemistry of LIMP-2 in MDCK monolayer culture. Endogenous LIMP-2 proteins (green, nuclear content blue) were found not only inside the cell but also enriched on the cell surface that formed borders between cells. **b**, **c** Time-lapse images of a MDCK cell transfected with a LIMP-2-HA construct (enlarged view in insets; time stamp shown in yellow). LIMP-2-HA located on the cell surface was visualized by surface labeling of mouse anti-HA antibody and anti-mouse Fab fragment conjugated to AlexaFluor488. A LIMP-2 containing endocytic vesicle (arrowhead) formed within 14 s was detected in **c** but not in **b**. **d** Western blotting with anti-LIMP-2 antibody. **e–g** Surface labeling of LIMP-2 in WT or KO MEF. **h** Quantification of surface labeling with LIMP-2 or IgG isotype antibody. **i–k** 30 min uptake of PC or PS liposomes in MEF. **i** WT MEF with few intracellular PC particles (arrow). **j** WT MEF with intracellular PS particles (green). **k** LIMP-2 KO MEF with few intracellular PS particles (arrow). **l** Quantification of liposome uptake in WT and LIMP-2 KO MEF. **m**, **n** Dyngo-2a (100 μM, 15 min) blocks both PS liposome (green) uptake and transferrin (red) endocytosis. **o** Nystatin (100 μM, 15 min) inhibits the endocytosis of PS liposome (green) but not transferrin (red). **p** Quantification of PS fluorescence intensity in WT MEF treated with endocytosis inhibitor Dyngo-2a or nystatin. **q–t** Endocytosed PS vesicles (green) were partially localized to lysosomes (lysotracker, red) after 60 min of uptake. **t** Enlarged view of the square in **s**. Results are expressed as means ± SE. *, $t$ test $P = 3.8\text{E}{-}4$; **, $t$ test $P < 2.2\text{E}{-}14$. Blue, DAPI. Scale bar: **a–k**, 12 μm; **m–o**, 25 μm; **q–t**, 12 μm

dimerization/multimerization of endogenously and heterologously expressed full-length LIMP-2 in HeLa cells in UV cross-linking experiments, native and denatured gels (Supplementary Fig. 10A–C, G, H). Co-immunoprecipitation demonstrated that these dimer/oligomers are LIMP-2 homodimer/homooligomers (Supplementary Fig. 10D, I). Like the purified LIMP-2 luminal domain dimer (Supplementary Fig. 3B), full-length LIMP-2 dimer migrated diffusely in native gels (Supplementary Fig. 10B, G). The diffuse nature of the LIMP-2 dimer band in native gel mostly likely reflects the heterogeneity of the bound lipids and ultimately limits the detection sensitivity of the endogenous LIMP-2 dimer. Nevertheless, endogenous LIMP-2 dimer was detected in wild-type mouse embryonic fibroblasts (MEF WT) (Supplementary Fig. 10B). Interestingly, a significant amount of endogenous LIMP-2 dimer was detected in Madin–Darby canine kidney epithelial cells (MDCK) under denaturing conditions (Supplementary Fig. 10E).

To directly visualize LIMP-2 dimer formation in live cells, we performed bimolecular fluorescence complementation experiments by tagging and expressing LIMP-2 with two complementary Venus non-fluorescent fragments (Fig. 4a). Dimerization of WT LIMP-2 resulted in strong green fluorescence signals in live cells that largely co-localized with the red fluorescence signals of co-transfected Lamp1, a major lysosomal marker (Fig. 4b), indicating correct trafficking of these LIMP-2 proteins to lysosomes. We also generated H150/F151D mutants tagged with the same complementary fragments. While the mutants expressed at similar levels as those of the WT LIMP-2 proteins (Fig. 4c and Supplementary Fig. 10F) and trafficked to lysosomes (Fig. 4b), they had significantly decreased dimer signals compared to the WT proteins (Fig. 4b). We further quantified the amounts of dimer formation of these proteins by calculating the ratios of LIMP-2 signals in lysosomes in each cell to the red fluorescence signals of mCherry targeted at endoplasmic reticulum in the same cell (Fig. 4d). In these experiments, fixed amounts of LIMP-2 and mCherry plasmids were used in co-transfections to control transfection variabilities. The amounts of LIMP-2 H150/F151A dimer were significantly less than those of the WT LIMP-2 dimer (Fig. 4d; WT, $0.55 \pm 0.03$, $n = 187$; H150/F151A, $0.326 \pm 0.018$, $n = 188$; $t$ test $P < 0.0001$), confirming a significant interference of these mutations with LIMP-2 dimerization.

**LIMP-2 mediates cellular uptake of PS liposomes.** Since the ectodomain of LIMP-2 can bind liposomes, we next investigated the lipid trafficking capability of LIMP-2 localized on cell surfaces. Small fractions of LIMP-2 was reported to be found on the plasma membrane of MEFs using a biotinylation assay[39]. Using immunocytochemistry, we found that a fraction of endogenous LIMP-2 is expressed on the surface of MDCK cells in confluent monolayers (Fig. 5a). When MDCK cells were transfected with a LIMP-2-HA construct and monitored with live imaging, anti-HA antibody bound to surface-expressed LIMP-2-HA was readily internalized into the cell, forming endocytic vesicles and trafficking inside the cell (Fig. 5b; time-lapse, 5c; Supplementary Movie 3). These results indicate that LIMP-2 can utilize the endocytic machinery to deliver its cargo from the cell surface to intracellular compartments. Specificity of the anti-LIMP-2 antibody used in this experiment was verified by immunoblotting (Fig. 5d and Supplementary Fig. 10J).

Using antibody labeling of live cells, LIMP-2 was detected on the surface of WT MEFs (Fig. 5e) and the mean fluorescence intensity was significantly higher compared to immunoglobulin isotype control or in LIMP-2 KO MEF (Fig. 5f–h; WT LIMP-2, $5950 \pm 135$; WT IgG, $3823 \pm 106$; $t$ test $P = 2.2\text{E}{-}14$). To visualize lipid uptake in MEFs, we used TopFluor-PS and TopFluor-PC

which have comparable fluorescence spectra that are insensitive to pH in the physiological range[40]. Within 30 min after adding liposomes, whereas PC fluorescent signal inside the cells was weak (Fig. 5i), PS-positive vesicles were readily detected inside WT MEF (Fig. 5j, l; fluorescence intensity mean PC, $2212 \pm 113$; PS, $8023 \pm 250$; $t$ test $P = 4.2\text{E}{-}30$). When using solubilized lipids to test uptake capacity, we found that 11-fold more PS particles than PC particles accumulated in WT MEFs (Supplementary Fig. 11). In addition, phospholipids were the preferred substrates for transport when compared to sphingolipids, such as ceramide and glucosylceramide (Supplementary Fig. 11). Consistent with our biophysical experimental results, these data suggest that PS is likely to be the dominant lipid species for cellular uptake. To determine whether LIMP-2 mediates liposome uptake, we measured PS liposome uptake in LIMP-2 KO MEFs (Fig. 5k). The mean PS fluorescence intensity in LIMP-2 KO cells was reduced to 44% of the WT level after 30 min uptake (Fig. 5l; WT, $8023 \pm 250$; KO, $3502 \pm 199$; $t$ test $P = 4.3\text{E}{-}20$). When the uptake time was extended to 60 min, while PS intensity increased in WT cells ($t$ test $P = 3.8\text{E}{-}4$), it did not change significantly in LIMP-2 KO cells, only reaching 38% that of WT (Fig. 5l; WT, $9160 \pm 308$; KO, $3473 \pm 183$). This result indicates that LIMP-2 contributes significantly to PS liposome uptake in cells.

Next, we investigated the transport route of LIMP-2-mediated liposome trafficking. Dynamin is large GTPase that mediates membrane scission during endocytic vesicle formation via clathrin-dependent and caveolae-dependent pathways. When cells were treated with the dynamin inhibitor Dyngo-2a (100 μM, 15 min) to block endocytic vesicle formation, we found that PS liposome uptake was completely inhibited in WT MEF (Fig. 5m, n, p; control, $9166 \pm 503$; Dyngo-2a, $668 \pm 36$; $t$ test $P = 8.7\text{E}{-}24$). PS liposome uptake can also be halted by blocking the caveolae pathway alone. When cells were treated with nystatin, which interferes with caveolae formation but not the generation of clathrin-coated vesicles[41], PS was sequestered on the cell surface, whereas transferrin was endocytosed through clathrin-coated vesicles in the cytosol (Fig. 5o, p; control, $9166 \pm 503$; nystatin, $1862 \pm 337$; $t$ test $P = 1.9\text{E}{-}18$). These results indicate that LIMP-2 can function as a PS receptor in MEF and mediate PS liposome uptake through dynamin-dependent endocytosis.

We further examined the subcellular localization of internalized PS. Within 1 h after adding PS liposomes to WT MEF, TopFluor-PS-positive vesicles exhibited a punctate distribution in the cytoplasm (Fig. 5q). A fraction of PS vesicles (22%) co-localized with the lysosomal marker LysoTracker (Fig. 5q–t; $N = 102$). These data indicate that a portion of the endocytosed liposomes are trafficked through endocytic vesicle compartments to lysosomes. Since LIMP-2 mediates the activation of GCase by transporting GCase from the ER to lysosomes where GCase enzymatically processes its sphingolipid substrates[7], we conducted three GCase activity assays to test whether liposome treatment interferes with cellular GCase activity. Treating WT MEF with PS or PC liposomes (1–3 h) did not significantly change GCase activity measured with a covalent probe MDW9[42], a fluorogenic GCase substrate 4-methylumbelliferyl β-D-glucopyranoside (4MUG), or 5-(pentafluorobenzoylamino) fluorescein di-β-D-glucopyranoside (PFBF) (Supplementary Fig. 12). These results indicate that LIMP-2 mediated PS liposome transport does not interfere with LIMP-2 mediated GCase activation.

## Discussion

In this study, we have defined a dimer structure of LIMP-2 with bound phospholipid and CLR molecules. We have demonstrated with multiple biophysical assays that the LIMP-2 luminal domain

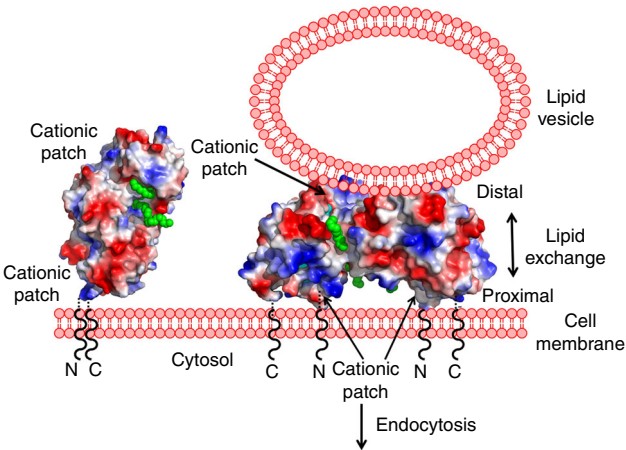

**Fig. 6** Proposed mechanism of lipids trafficking by LIMP-2. Lipids (green spheres) binding of LIMP-2 monomer promote dimer formation. In the LIMP-2 dimer the luminal domains are closer to the cell membrane, with the proximal face of LIMP-2 luminal domain able to interact with the cell membrane while its distal face able to interact with extracellular lipid vesicles, respectively, via their cationic patches. Within the tethers of the protein on the luminal side, between the ends of the structurally defined polypeptides and the transmembrane helices, there are sufficient residues that are flexible and lack defined secondary structure to enable this dimerization on cell surface: 24 residues after the last residue of the N-terminal transmembrane helix and 6 before the beginning of the C-terminal transmembrane helix. If fully extended, these tethers can stretch at least 20 Å. Notice the channels in LIMP-2 ectodomain dimer connect bound lipid vesicles and cell membrane, with openings near cationic patches in both faces that may facilitate lipids exchange between vesicles and outer leaflet of the cell membrane. LIMP-2 bound lipid vesicles can be trafficked into cell cytosol via endocytosis

binds phospholipids and characterized the distinct binding capabilities of LIMP-2 when isolated in monomeric or dimeric forms. Notably, the dimer binds and tethers liposomes much more readily than the monomer, with a strong preference for PS liposomes. Full-length LIMP-2 dimer can be detected in MEF, HeLa, and MDCK cells. With cell uptake assays, we have shown that LIMP-2 preferentially mediates the uptake of PS liposomes through dynamin-dependent endocytosis. Internalized PS liposomes are partly delivered to the lysosome, where GCase was determined to remain active in different cellular assays.

The dimerization of LIMP-2 is mediated by protein–protein interactions, interactions involving the bound phospholipids and glycans at N325. In a previous study, extensive mutations on the α5 and α7 did not disrupt dimerization of the ectodomain[23], and in this study, a minimal number of point mutations at the observed dimerization interface disrupted the dimer formation to a limited but significant degree. Similar physiological dimers were reported for SR-B1[43–45] and CD36[27, 46]. Our structure explains why lipid binding favors LIMP-2 dimerization and shows how lipid-binding converts the GCase-binding LIMP-2 monomer to a dimer that is able to tether lipid bilayers.

Since GCase was shown to bind PS-containing liposomes with similar affinity[47] as LIMP-2 monomer, it is likely that GCase can also be sorted to lysosomes as cargo of LIMP-2 bound lipid vesicles. That may explain why the addition of liposomes did not impede GCase function in cells. Alternatively, LIMP-2 may act as a dual receptor for both surface vesicle internalization and intracellular GCase trafficking. Similarly, LIMP-2 may also facilitate enterovirus 71 entry into cells as cargo of endocytic vesicles, in addition to its proposed direct interactions[25]. Interestingly, it was reported recently that clustered packaging of

enterovirus within PS enriched vesicles infects cells more efficiently than free viruses[48]. Further studies are needed to differentiate these two trafficking routes.

The impacts of H150A/F151D mutations on LIMP-2 dimerization in cells and modeling studies indicated that the LIMP-2 luminal domain dimer observed in the crystal structure is able to form on the cell surface (Fig. 6). The ability of the LIMP-2 luminal domain dimer to tether liposomes suggests that it may simultaneously interact with the cell membrane and with bound extracellular lipid vesicles, most likely via its proximal face and its distal face, respectively, when the transmembrane segments of the full-length LIMP-2 are anchored in the cell membrane (Fig. 6). This potential bridging of cell membrane and lipid vesicles by the LIMP-2 luminal domain may have biological implications: the internal CLR sites and the external PC clefts may serve as channels for lipid exchange. Mutagenesis and covalent inhibitor labeling[23, 33, 34, 37] suggest that SR-B1 and CD36 may use equivalent sites in their protein dimers for the selective uptake of CLR, FA, and phospholipids. Conformational switching of class B scavenger receptors may provide a general mechanism for regulated membrane association and lipid trafficking. Taken together, our results define a lipid-regulated dimeric state of LIMP-2 that serves as a molecular switch between GCase transport and lipid trafficking with implications for cellular homeostasis and neurodegenerative disease.

## Methods

**Protein expression and purification.** LIMP-2 was expressed using high-density mammalian transient protein expression in Expi293F™ (Human Embryonic Kidney cells, HEK) cells (ThermoFisher Scientific) and N-acetyl-glucosaminyltransferase I deficient HEK 293GnTi– cells (ATCC® CRL-3022) using ExpiFectamine™ 293 reagent (ThermoFisher Scientific). The construct designed included the soluble luminal domain of LIMP-2, residues 28–431 with an N-terminal Kozak sequence and IgK signal peptide for secretion, and a C-terminal 6xHis-tag. The code-optimized DNA was synthesized and inserted into pUC57-Kan vector (Genewiz). It was subcloned using restriction sites BamHI and NotI into a pcDNA3.1Hygro(+) vector (Thermo Scientific). The cells were incubated post-transfection for up to 60 h before the cells were pelleted by centrifugation and the soluble fraction was filtered at 0.22 μm. Media was dialyzed overnight in DPBS. The protein was His-tag affinity purified by binding to Talon (Clontech) or HisPur resin (ThermoFisher Scientific), washing with 10 mM imidazole, 150 mM NaCl, 10 mM HEPES pH 7.5 and eluted with 500 mM imidazole, 150 mM NaCl, 10 mM HEPES pH 7.5. The purified protein was further purified on size exclusion (Superdex 200, GE) using a running buffer of 10 mM HEPES pH 7.5, 150 mM NaCl, HIC (HiTrap Phenyl HP, GE) with buffer of 10 mM HEPES pH 7.5, 150 mM NaCl, and ion exchange chromatography (HiTrap Q, GE). The buffer for ion exchange contains 10 mM HEPES, pH 7.5, 50–1000 mM NaCl, and fractions pooled were used directly for crystallization after concentration.

F151D and H150A mutants were generated using Q5 mutagenesis kits (New England BioLabs) (see Supplementary Table 3 for primers used). These mutants were expressed in Expi293F(HEK) and purified in the same manner as the wild-type protein.

GCase was expressed in Sf9 cells using the Bac-to-Bac Baculovirus expression system (ThermoFisher Scientific) using synthesized DNA construct. The construct included the native 39 residue signal peptide and no additional affinity tags. Cells were infected at a cell density of 2 × 10^6 with amplified virus and harvested after 4 to 5 days. Cells were pelleted by centrifugation, the soluble fraction was filtered at 0.22 μm, and subsequently purified by HIC (Toyo Butyl Pearl) using a washing buffer of 20 mM sodium acetate pH 5.0, 600 mM NaCl, and an elution buffer of 20 mM sodium acetate pH 5.0, 150 mM NaCl and 50% ethylene glycol, followed by Heparin purification (HiTrap Heparin, GE) eluted with a salt gradient of 50–1000 mM NaCl, in buffer of 20 mM sodium acetate pH 5.0 and 20% ethylene glycol. The purified protein was further purified on a SEC column (Superdex 200) using running buffer of 10 mM MES pH 6.0, 100 mM NaCl, 1 mM TCEP, and deglycosylated with PNGase F (New England BioLabs).

The C-terminal BAP-tagged constructs of LIMP-2 and GCase used in binding experiments were generated using PCR (Supplementary Table 3) and were expressed in Expi293F and purified as described above.

**Crystallization.** The protein was thoroughly screened in a number of crystallization conditions; included in select screens was additive compounds and/or GCase. Crystal trays were set up using a Mosquito crystallization robot, with sitting drops that underwent vapor diffusion composed of 0.2 μL of condition mixed with 0.2 μL of protein. The protein was a mix of 30 mg mL−1 of equal amount of LIMP-2

and β-GCase. The drop that produced the highest-resolution diffracting crystals was from an additive screen in which 30 mM Glycyl–glycyl–glycine was added to the well condition of 0.221 M NaCl, 29.2% w/v PPG P 400.

**Data collection and structure determination.** A 3.0 Å data set was collected at IMCA17ID beamline (APS, Chicago), at a wavelength of 1.0 Å and temperature of 100 K, from a single crystal flash cooled in liquid nitrogen. X-ray data was processed using program XDS[49]. The structure was determined using molecular replacement method using apo LIMP-2 structure (PDB code 4Q4F) as starting model. Structure was refined using program Buster[50] and manual model building using Coot[51]. Ramachandran statistics are 99% in preferred and allowed regions with 1% outliers. All outliers are in flexible loops. The overall MolProbity score is 1.88 and clash score is 0.82.

**Liquid chromatography–mass spectrometry.** The phospholipid content of LIMP-2 samples was determined by LC–MS[52–54] performed on an Agilent 1100 capillary system interfaced with an LTQ Orbitrap XL spectrometer (Thermo-Fisher). An isocratic separation was performed with the LTQ Orbitrap tuned to detect 2-oleoyl-1-palmitoyl PC dissolved in methanol at 5 mg mL$^{-1}$. Fractionation was performed using a $150 \times 0.5$ mm Zorbax C18 column (Agilent) (5 μm particle, 80 Å pore size) fitted with a $10 \times 1$ mm Vydac C18 guard column (Grace) (5 μm particle) at 40 °C using isocratic elution at 5 μL min$^{-1}$ with 65% methanol, 35% acetonitrile, 0.6% formic acid, 0.07% NH4OH. Mass spectra from $m/z = 400-2000$ in positive-ion mode were acquired in the LTQ Orbitrap XL, with the most abundant ions targeted for concurrent MS/MS in the linear ion trap with collision energy of 35%. The isolation width was set to 2 Da, and dynamic exclusion of precursor ions was activated after 3 scans within 30 s. Instrument tuning was performed using a solution of 2-oleoyl-1-palmitoyl PC (Sigma) dissolved in methanol at 5 mg mL$^{-1}$ (6.6 mM) and diluted 1:1000.

**Liposome generation.** Liposomes were prepared according to the Avanti Polar Lipids, Inc procedure (https://avantilipids.com/tech-support/liposome-preparation/). Lipids were purchased from Avanti Polar Lipids, Inc, dried by dry vacuum if chloroform was present, solubilized in 10 mM HEPES pH 7.5, 150 mM NaCl buffer at concentration minimum of 1 mg mL$^{-1}$, sonicated/vortexed for ~30 min, followed by 10 freeze-thaw cycles in liquid nitrogen and warm water with vortexing upon thawing, and 100 nm extrusion in a Avanti mini-extruder. Extruded liposomes were utilized for a maximum of 3 days and stored at 4°. The fluorescent liposomes used for cell assays were 1 mM in PBS (pH 7.2). The liposomes contained 10% TopFluor (Avanti) lipid (TopFluor PC, 810281; TopFluor-PS, 810283), along with the corresponding unlabeled phospholipids, and were prepared as described with extrusion at 100 nm.

**SPR.** SPR experiments were performed on a Biacore 3000 (GE). Experiments were designed to monitor the binding of LIMP-2 in the mobile phase to liposomes immobilized on a Biacore L1 chip surface (Pioneer), and the reversely, liposomes in the mobile phase to LIMP-2 immobilized via biotin tag to neutravidin captured on a CM5 chip (GE). To immobilize liposomes on the surface of the L1 chip, the chip was conditioned with two injections of 100 mM NaOH prior to liposome binding. Liposomes were injected onto the L1 surface at a rate of 5 μL min$^{-1}$ to a range of 2000–4500 RU. The adhered liposomes were washed with 2 injections of 100 mM NaOH to remove loosely adhered phospholipid, or removed from the chip by 20 mM CHAPS. LIMP-2 in both primarily monomeric and primarily dimeric forms isolated from size exclusion and further purified individually by ion exchange chromatography were injected at a flow rate of 50 μL min$^{-1}$ at concentrations ranging from 12.3 nM to 18 μM, in either twofold or threefold dilution series. Methods included either buffer washes after analyte injections and/or long dissociation times of 600–1000 s. BSA was included in the running buffer at 0.1 mg mL$^{-1}$ to minimize nonspecific interactions. To test LIMP-2 binding at different pH, buffers of 10 mM HEPES pH = 7.5, 150 mM or 10 mM NaAcetate pH = 5.0 or 5.5, 150 mM NaCl were used.

To generate immobilized LIMP-2 or GCase, the C-terminal BAP-tagged construct was enzymatically biotinylated with a BirA biotinylation kit (Avidity). Using a CM5 chip (GE), the surface was activated for amine coupling of neutravidin with EDC/NHS, neutralized following neutravidin injection with ethanolamine, and conditioned with SA rinse according to established protocol (GE). LIMP-2 was immobilized to 200–1000 RU for experiments of protein and liposome interactions. Removal of bound lipid was completed with rinses of 20 mM CHAPS. GCase was immobilized to 1400 RU for studies of protein–protein interactions. LDL particles (Kalen Biomedical, LLC) were immobilized similarly.

**Multi-angle static light scattering.** Multi-angle static light scattering experiments were completed on a Wyatt Technology light scattering instrument with a Dawn Heleos-II detector and Optilab T-rEX dRI detector, in conjugation with an 1220 Infinity LC (Agilent technologies) with a Shodex KW404-4F semi-micro SEC column. A total of 50 μg of protein was analyzed in either PBS pH 7.2 or 100 mM sodium citrate in PBS at pH 5.0, diluted from concentrated stocks in their respective purification buffers.

**DLS.** DLS profiles were collected on a Wyatt DynaPro Plate Reader in a 384-well plate. Sample concentrations were 0.25 mg mL$^{-1}$ each for both lipids and proteins (~4 μM protein concentrations) in 10 mM HEPES pH 7.5, 150 mM NaCl. Readings were completed 15 times for increments of 5 s, in duplicate.

**Negative stain EM.** Square mesh thin carbon-coated copper EM grids (CF300-CU-UL, Electron Microscopy Sciences) were prepared for negative stain EM images. Uranyl formate (UF) was prepared as a 1% w/v solution in deionized water and protected from light. The solution was filtered at 0.02 μm before use. Samples were diluted in 10 mM HEPES pH 7.5, 150 mM NaCl to 0.125 mg mL$^{-1}$ (protein and lipids each). An aliquot of 4 μL of sample was added to a glow discharged grid, positioned over ice. The sample was then blotted for 1 or 2 min with Whatman filter paper. The blotted grid was then touched to a 20 μL well of UF for 30 s, then blotted with filter paper, followed by an additional UF touching/blotting step for 1 min. Drying of the grid was initiated with a slow stream of N$_2$ gas for approximately 30 s, then the grids were left overnight to dry completely on filter paper in a covered petri dish before storing. Negative stain EM images were collected on a FEI Technai TF200 microscope operated at 200 kV. Images were collected at 1700 and ×6500 magnification using a Gatan US1000 2k × 2k CCD camera.

**Cross-linking experiments.** Murine wild-type LIMP-2 cDNA was cloned into pFrog-Vector (a derivative of pcDNA3.1) as described[7]. To generate the concatemer the murine LIMP-2 cDNA was fused to the c-terminus of LIMP-2 using PCR techniques and restriction enzymes. Both LIMP-2 constructs were C-terminally myc-tagged and verified by sequencing. Untransfected or transient-transfected Hela cells were washed once with PBS and twice with label medium (DMEM minus leucine and methionine, 10% dialyzed FCS, Pen/Strep), following incubation for 24 h with label medium containing pac-methionine (1.5 mM) and pac-leucine (3 mM). After labeling, cells were washed with PBS and exposed to UV light (365 nm) for 15 min according to the manufacturer's protocol (Thermo-Fisher). Cells were harvested, lysed and analyzed via SDS–PAGE and immunoblot.

**Electrophoresis and immunoblotting.** HeLa cells were grown to 70% confluency in DMEM + 10% FCS + 1% P/S and transfected with murine LIMP-2-myc for 48 h. Transfected and untransfected HeLa cells, WT MEFs, LIMP-2$^{-/-}$ MEFs, WT MDCK cells were harvested, lysed, and analyzed with the Native PAGE™ Bis-Tris Gel System (ThermoFisher) according to the manufacturer's instructions, or via SDS–PAGE, and immunoblotting. For co-immunoprecipation, HeLa cells were transiently transfected with GFP or co-transfected with two differently tagged LIMP-2 constructs (LIMP-2-myc and LIMP-2-HA) for 48 h. Cells were harvested, lysed and the lysate was incubated with anti-myc antibody (9B11, Cell Signaling) overnight at 4 °C. Then, magnetic agarose G beads (ThermoFisher) previously blocked with 3% BSA were added to the samples (40 min, 4 °C, rotating) for the precipitation of LIMP-2-myc. Afterwards, samples were analyzed via SDS–PAGE with reducing reagents and immunoblotting. An anti-HA-tag antibody (3F10, Roche) was used to detect co-precipitated LIMP-2-HA.

**Bimolecular fluorescence complementation.** HELA cells were cultured in DMEM supplemented with 10% FBS (v/v). The cells were plated into each chamber of a 4-chamber, glass bottom, 35 mm dishes (Cellvis) at a density of 60,000 cells per well. After 24 h, the cells were transfected (Lipofectamine 3000, Invitrogen) with 125 ng of LIMP-2 (WT or H150A/Y151D) fused with the N-terminal and C-terminal Venus fragments (pBifC-VN(I152L), Addgene plasmid # 55041 and pBiFC-VC155, Addgene plasmid # 22011, gifts from Chang-Deng Hu[55]). To control for transfection variability 50 ng of ER targeted mCherry (Addgene plasmid # 55041, a gift from Michael Davidson) was included in the transfection. To confirm lysosomal localization of the LIMP-2 constructs, additional cultures were transfected with Lamp1-RFP (Addgene plasmid # 1817, a gift from Wather Mothes[56]). After 12 h cells were imaged using a spinning disk confocal microscope (Nikon). Quantification of BiFC efficiency was carried out using the NIS Elements software (Nikon). Regions of interest (ROI) were drawn around Venus and mCherry positive cells using differential interference contrast view. The mean fluorescence intensity normalized to ROI surface area was determined and used to generate a ratio of Venus/mRFP. The Venus/mRFP ratio was determined from >180 cells (~60 cells from 3 independent replicates). To ensure that the BiFC constructs were expressed at equal levels, after imaging, the cells were lysed, combined with Laemmli buffer and subject to western blot analysis. Membranes were probed with primary antibody specific to LIMP-2-VC-HA (#3724, Cell Signalling, 1:1000) or LIMP-2-VN-Myc-tag (#2276, Cell Signalling, 1:1000); and anti-mouse IR-Dye 680 (Licor, 1:10,000) or anti-rabbit ID-Dye 800 (Licor, 1:10,000) secondary antibodies. Bands were visualized with an Odyssey CLx near-infrared fluorescence imaging system (Licor).

**Cell culture and immunocytochemistry.** MDCK cell line (ATCC) was maintained in DMEM plus 10% FBS. When MDCK cells were reaching confluence, we switched culture medium to DMEM plus 0.2% FBS and allowed MDCK monolayer to grow for another 2–3 days before conducting experiments. For immunocytochemistry, cells were fixed with 100% methanol for 12 min and stained with anti-LIMP-2 antibody (Abcam ab176317, 1:50 dilution). Wild-type and LIMP-2$^{-/-}$ MEF were maintained in DMEM medium plus 10% FBS. For live cell surface

labeling, MEF were incubated with an anti-LIMP-2 ectodomain antibody (LifeSpan Biosciences, 1:50 dilution) or rabbit IgG isotype control (ThermoFisher) for 15 min at 37 °C, rinsed off excess antibody, fixed with 4% paraformaldehyde (PFA), and stained with secondary antibody. Confocal Z-stack images were taken with LSM 880 (Zeiss) under the same optimal acquisition setting and quantified with ZEN software (Zeiss) without applying any intensity threshold limit. We delineated the border of individual non-overlapping cell to measure the pixel intensity mean value within the region and then subtract the background intensity value. Background pixel intensity was derived from adjacent fields that had no cell presence. Images of each condition were compiled from at least 3 independent experiments and the measurements of 30–60 cells per condition were used for statistical analysis.

**Time-lapse imaging.** MDCK cells were transfected (Lipofectamine 2000, Invitrogen) with a LIMP-2-HA construct which was generated by inserting a self-annealed double strand DNA duplex containing a HA tag in the ectodomain (position A299) (Supplementary Table 3) of a LIMP-2 full-length expression construct driven by CMV promoter. Live cell imaging and surface labeling of LIMP-2-HA was conducted as described by Kennedy et al.[57] with slight modification. Briefly, cells were first incubated for 3 min in mouse anti-HA antibody (Covance, 16B12) and then 3 min in anti-mouse IgG F(ab′)$_2$ fragment conjugated to AlexaFluor 488 (Jackson Immuno). Imaging commenced immediately after washing off the secondary antibody with a spinning disc confocal microscope AxioObserver.Z1 (Zeiss) at 37 °C.

**Phospholipid uptake assay.** MEF cells were plated at a density of $2.5 \times 10^4$ cells cm$^{-3}$. Three days after seeding, cells were treated with 5 μM liposomes, 5 μM solubilized lipids, or 20 μM exosomes diluted in DMEM medium with HEPES (pH7.4) and incubated at 37 °C for 30 to 60 min. BSA-solubilized lipids were made up of 10 mM lipid in 12% w/v BSA with 150 mM NaCl and solubilized by vortexing[58]. In total, 100 nM Lysotracker-Red/Deep Red (Invitrogen) or 5 mg mL$^{-1}$ transferrin-Alexa594/Alexa488 (Invitrogen) were mixed with the lipid uptake solution if needed. For endocytosis inhibition, cells were pretreated with 100 μM of Dyngo-2a (Abcam) or nystatin (Sigma) for 15 min at 37 °C before adding in the liposome mixture. Cells were rinsed with PBS to wash off unbound reagents and fixed with 4% PFA. Mounted slides were stored at 4 °C and confocal microscopy images were taken with LSM 880 (Zeiss) within 5 days post-fixation. For pixel intensity measurement, images quantification was conducted as described in the cell culture and immunocytochemistry section. For lysosomal colocalization, z-stack confocal images were acquired under the optimal acquisition setting of LSM 880 (Zeiss). Built-in Coloc function in the Imaris software (Bitplane) was used for volume rendering, spot detection, and overlapping percentage count.

**Synthesis of MDW941.** The synthesis was modified from the approaches described by Madsen and Overkleeft in their syntheses of cyclophellitol[59] and MDW941[42], respectively. The route was optimized for a single compound (MDW941) resulting in a shortened synthetic sequence (14 → 12 steps, longest linear sequence) (Supplementary Methods). This was achieved by circumventing a protecting group exchange as described in the initial disclosure[42]. Further optimizations to the experimental procedures resulted in enhanced material throughput. Detailed synthesis scheme was outlined in Supplementary Fig. 13.

**GCase activity assays.** Live cells were pretreated with liposomes for 1 h as described in the phospholipid uptake assay before adding GCase activity detection reagents. 400 nM MDW941 was added into the liposome uptake solution and further incubated for 30 min at 37 °C. Cells were rinsed with PBS and fixed with 4% PFA. For 4MUG (Sigma) reaction, 1.25 mM 4MUG was diluted in pH 4.0 acetate buffer and incubated with the cells for 1 h at 37 °C. GCase activity was stopped by basifying the reaction with 3× volume of 0.2 M glycine pH 10.8 solution. For PFBF (Molecular Probes) readout, cells were pretreated with GCase inhibitor conduritol B epoxide (CBE, Sigma) for 15 min, followed by liposome treatment for 1 h, and then incubated in 0.5 mM PFBF for 30 min. Each condition was done in triplicate. Fluorescent signals of activated 4MUG and PFBF were detected with Envision plate reader (PerkinElmer).

**Data availability.** Coordinates and structure factors have been deposited in the Protein Data Bank with accession code 5UPH. Other supporting data are available from the corresponding authors upon reasonable request.

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

## Acknowledgements

We thank Dr Herman Overkleeft and Dr Johannes Aerts for providing their batch of MDW941 probe. We thank Katherine Hales for her assistance in image analysis, Laura J. Byrnes for assistance with negative stain EM, Johann Groth for performing in cellulo cross-linking and Kieran G. Geoghegan for proofreading the manuscript. This work was supported through grants from the Deutsche Forschungsgemeinschaft to P.S. (GRK1459) and M.S. (GRK1459 and a Heisenberg Fellowship), grant from US NIH to D.K. (R01NS076054).

## Author contributions

K.S.C., T.-W.C. and S.L. designed experiments and analyzed data. K.S.C. made LIMP-2 luminal domain constructs, purified proteins, determined crystal structure, prepared liposomes and performed biophysical experiments. T.-W.C. conducted all cell activity experiments. D.Y., S.H. and K.S.C. performed full-length LIMP-2 monomer and dimer detections in cells. K.S.C. and S.L. analyzed the structure. L.R.H. performed LC/MS and MS/MS experiments and analyzed the data. M.S. supervised in cellulo cross-linking experiments. D.K. supervised D.Y. B.A.C. helped analysis of SPR data. C.W.a.E. synthesized the probe MDW941. P.S. provided LIMP-2 KO mice and MEF. P.S. and M.S. helped to plan experiments. S.L., Q.X. and M.D.E. supervised experiments. All authors contributed to manuscript preparation.

## Additional information

**Competing interests:** The authors declare no competing financial interests.

