## [Peer Review File · Nature Communications]

Reviewers' comments:

Reviewer #1 (Remarks to the Author):

Comments to the authors:

The key new findings reported in this manuscript are:

- 1) The structure of the dimeric form of LIMP-2 luminal domain – this result may have important implications for other class B scavenger receptors (CD36 and SR-BI). Note: the authors should clearly indicate in the abstract and elsewhere that they do NOT determine the structure of the whole LIMP-2, but only its luminal domain.
- 2) The monomeric and 'dimeric' (possibly oligomeric) forms of the LIMP-2 luminal domain exhibit distinct binding properties – monomer preferred for binding GCCase whereas 'dimer' preferred for phosphatidyl serine (PS). Again, a very intriguing result.
- 3) Possible modes of binding of lipids to hydrophobic tunnel in LIMP-2 (possible, because the figures were not sufficiently clear to convincingly establish the nature of that binding)

Most of the structure figures need to be redrawn to better show the results they claim they have (see details below). For example, it will be necessary to show clearly the electron density for the lipids bound to the LIMP-2 and clearly show and describe how those lipids are related to the hydrophobic tunnel and protein that lines the tunnel. This is a key issue that they did not address adequately. Also, they need to better show the dimer interface and the hydrophobic cleft they say is generated by dimerization. The structural information is one of the two key types of findings and it has to be presented in a form better accessible to the reader.

The second key finding – that LIMP-2 can serve as a receptor for PS – is definitely worth reporting. However, the authors seem unaware that others have previously reported that class B scavenger receptors (CD36, SR-BI) can function as PS receptors. The first report of this that I could find was Rigotti et al. *J Biol Chem.* 1995 Jul 7;270(27):16221-4. Given the strong emphasis the authors make on this activity of LIMP-2, it seems worthwhile to put it in the context of earlier related (though not identical) work.

The authors take considerable space describing experiments that tell us what we already know – that LIMP-2 moves bound molecules to the lysosome where GCCase (one of its ligands) functions. All of those studies (eg., Fig4M-T) could be moved to the supplement, as they do not provide new insights. There are also some problems with interpreting some of the data, e.g., some cells do and some don't take up exosomes, etc.. These panels are substantially less impressive or significant than the data in the rest of the manuscript. The authors should consider leaving these data out of the manuscript. It seems that these are non-definitive data that reduce, not increase, the impact of the manuscript.

The authors take considerable space in the results and abstract describing LIMP-2's distribution in the kidney and speculating about its function. However, their own data suggest that the kidney defects they describe in older LIMP-2 KO mice are not seen in younger LIMP-2 KO mice, and thus it is not the absence of LIMP-2 per se, but rather secondary consequences of its absence over time that are responsible for some of the defects they report. Interpreting these findings is problematic and they seem to reduce the impact of the work. The authors may want to consider removing the kidney findings from this manuscript and reporting them later when they have a more definitive story to tell.

Specific comments:

TITLE: The title is a bit misleading: "Lipids-bound structure reveals lysosomal integral membrane protein-2 as a phospholipid receptor" It was a series of binding studies in vitro and in vivo that established that LIMP-2 is a phospholipid receptor. The structure did NOT reveal it to be a phospholipid receptor, but rather to show together with MS analysis that there were phospholipids associated with their purified luminal domain. The authors may want to consider revising the title.

INTRODUCTION:

PLEASE GIVE MORE EXPLANATION FOR THE FOLLOWING: "Deficient lysosomal sorting of GCase due to LIMP-2 mutations is unlikely to be the primary cause of AMRF." NOT CLEAR TO THE READER.

P3 – please cite primary papers that are relevant. For example ref #14 is not appropriate: "LIMP-2 belongs to the class B scavenger receptor family that also includes CD36 and SR-B1. CD36 and SR-B1 are primarily receptors for fatty-acids and high-density lipoproteins (HDL), respectively, but their known ligands have expanded to include collagen, low-density lipoproteins (LDL), and phospholipids¹⁴." Reference 14 is not the proper reference for the CD36 and SR-B1 activities.

P4: "we establish that LIMP-2 binds phospholipids" data for other SRBs? Citation?

RESULTS

P5. The authors must explicitly state very early in the manuscript that they do not have the entire structure with transmembrane and cytoplasmic domains.

The authors point out that "important differences from previous LIMP-2 structures^{14,17,18} were evident in a lipid-bound structure of the LIMP-2 luminal domain". This seems to be a key point and it would be nice if clear figures showing these differences would be presented. Also, structure of CD36 extracellular domain should also be included in these comparisons.

P5 They should also state in the text the resolution of the structure and a measure of the quality of the data and the fit of the model. THIS IS ESSENTIAL. This should not be hidden in a supplemental table.

P5. They correctly state that "Two important differences from previous LIMP-2 structures^{14,17,18} were evident" "Second, clear densities for lipids, presumably acquired from the expression host, were present within each monomer at the hydrophobic tunnel (phosphatidylcholine, PC; cholesterol, CHO. Fig. 1B and Supplementary Fig. S1)." However, it would be appropriate to state here that lipids in the tunnel have been observed in the CD36 structure. (Nat Commun. 2016 Sep 26;7:12837.)

P5 Fig 1 and Sup Fig 1 – These figures do NOT clearly present the relationship between lipids and the hydrophobic tunnel. At least one additional figure clearly showing the lipids with their electron densities and showing them in the envelop of the tunnel and their relationships to the overall structure of the extracellular domain are essential. Also, what are the relationships of the side chains and backbone of the tunnel to the different portions of the lipids (e.g., alcohol in cholesterol, head groups of the lipids. How good are the fits of the electron density to the models of the lipids, particularly that of phospholipids and cholesterol? Are there alternative for the molecules described as lipids (e.g. polypropylene glycol)? These must be shown clearly and described precisely. On approach to this is seen in the figures in the CD36 paper (Nat Commun. 2016 Sep 26;7:12837.).

Figure 1: The labeling of the lipids in the surface models is very confusing – it is very hard to know what is the protein and what is the lipid – a better way of presenting the lipids than the thick block letters ("CHO", "PC") must be used so that it is easy for the reader to see where the lipids are located in these views. Are these lipids really accessible at the surfaces and from each orientation? Also, if the authors showed the relative orientations of the views in 1C, 1D and 1E using arrows and rotations showing the degrees, that would help a reader.

Figure 1A – please label the N and C termini of the two proteins. Also, what is the orientation of 1A relative to 1C,D,E? It seems like the same as in 1C, but not explicitly stated.

P5. What is the basis for the following statement: "Both faces have positive charge patches next to

hydrophobic areas, a combination that favors interaction with phospholipid bilayers." Again, a clear figure would help.

P5 – Exactly where are the hydrophobic clefts? This is not clear from Fig 1. "but dimerization combines these features and forms the additional hydrophobic clefts." This is one of several cases in which the figures are inadequate in presenting the structures.

P6 and Sup Fig 2A What is the basis of calling the area (left panel) indicated as 'dimer' as a dimer and not higher order oligomers? Why is the 'dimer' such a broad peak? [Indeed panel 2B suggests that there are higher order oligomers!] Why are the elution volumes so different in the right and left panels of Fig S2A? Define "SEC". Why does so much of the complex after SDS/reduction run as an apparent dimer? Do you see covalent cross-links in the crystal structure? Please explain the abbreviations in Fig S2C. FigS2D – what about the amounts of cholesterol?

FigS2D – quantitation by LC/MS without internal standards is a very iffy matter. The authors present the reduction in lipid mass in the monomer mutants without any comment about the limitations of such measurements. Also, in the legend they mention some samples of monomer had no lipid, but they don't indicate if this was a function of different preparations of protein samples or what exactly was the difference between different samples. This reviewer needs more before the claim in the legend can be considered justified. The data and statements are unconvincing.

P5: "The dimeric structure defined by crystallography concurred with results from multi-angle light scattering (MALS, Supplementary Fig. S2C), structure-guided mutagenesis (Supplementary Fig.S2A), and ligand content analysis (Supplementary Fig. S2D)." Please explain how the ligand content analysis results concur with the dimeric structure.

FigS3A – please define MOCK. Why was there so much LIMP2 there? Mock usually means mock transfected, but perhaps it means no photoactive amino acids? Experimental details should be provided so that a reader understands what has been done.

P6: "Modeling studies indicated that the LIMP-2 luminal domain dimer is able to form and to interact with the cell surface when its transmembrane segments are anchored in the cell membrane (Supplementary Fig. S3B)." What is the significance of the dimer model being able "to interact with the cell surface"?

P6: "The PC binding site is at the entrance half to the hydrophobic tunnel (Fig.1A), with the head group pointed inward (Fig. 2A)." Neither Fig1A or 2A clearly shows the relationship of the PC to the hydrophobic tunnel – please provide a more helpful image. Also regarding "Near the PC head group, D272 and helix α -9 that connects strands β 10 and β 11 are flipped out of the channel to accommodate and anchor the quaternary amine of PC", it would be helpful to have a figure that shows this more clearly, including appropriate labeling.

P6 and Fig 1A. The figure does NOT adequately give the reader a feel for how cholesterol is bound: "The CHO molecule is located in the lower half of the hydrophobic tunnel (Fig. 1A)." An image showing how the cholesterol fits the electron density must be shown as well as its relationship to the tunnel and the residues lining the tunnel. The issue of how cholesterol might fit into SR-B tunnels came up in the initial LIMP-2 structure paper and these authors appear to be in a position to provide a definitive answer. Unfortunately, the figures shown do not provide an adequate demonstration of the quality of the fit of the cholesterol model to the electron density nor its relationship to the tunnel and the LIMP-2 residues surrounding it. Again, the authors claim that somehow Fig 2B shows the following: "To accommodate CHO, the loop connecting β 9 and β 10 flips away and is entirely displaced by the α -helical bundle of the opposite subunit (Fig. 2B). Additional peptides are disordered/displaced at or near the CHO binding site, including α 1, β 4 – β 5, β 8 – β 9, and α 10 through α 11 (Fig. 2B). The displacements/disorder of α 1, α 10 and α 11 which cover the

CHO site in the monomer structures makes the CHO binding site exposed to both the distal and the proximal faces (Fig. 1C and 1D).” However the loop etc are not even labeled and how can one know what changes occurred when there is no clear comparison of the bound and unbound forms. The exposure of the CHO (not a normal abbreviation for cholesterol, but is for carbohydrate!) and/or its binding site is not at all clear This is yet again an issue where, one presumes, the authors have strong evidence supporting their claim, but they do not show the data (structures) in a way that provides firm support for the claim. And again, on P7 the authors refer to a figure that does NOT clearly illustrate what they say: “At the LIMP-2 dimer interface, the SN2 acyl chains of PC molecules and the glycans on N325 are involved in LIMP-2 interactions (Fig. 2D).”. This reviewer URGES the authors to reconsider how they are representing the structure so that a reader will be able to understand (and be convinced by) how the authors draw their conclusions. Otherwise, given the figures provided it can only be taken on faith that much of what they say is supported by their structure. [Note: FigS11B&C ARE good representations that allow the reader to understand what the authors are saying.]

P6: “In SR-BI, mutagenesis and covalent inhibitor labeling indicated that the equivalent CHO site is critical for the selective uptake of CHO14,25,26.” Please indicate which residues that are in LIMP-2 are equivalent to those in SR-BI are under consideration. Also, while the three citations given are relevant, the correct citations for the mutagenesis of SR-BI by Yu et al are: *Biochemistry*. 2012 Dec 18;51(50):10044-55; and *Proc Natl Acad Sci U S A*. 2011 Jul 26;108(30):12243-8.

P6: What was the rationale for placing a PPG fragment instead of an acyl chain fragment (or even a water) in the density described: “A fragment of polypropylene glycol precipitant (PPG) was placed in one of the clefts of the LIMP-2 dimer (Supplementary Fig. S1).” Please make the logic clear.

P7: The authors appear to make what may be a very important observation about the role of dimerization of SR-B protein in ligand binding, with special reference to CD36 and SR-BI: “This site along $\alpha 5$ is known to be involved in HDL binding for SR-B1 and oxidized LDL binding for CD-3614, but it is completely formed only in the dimer form of LIMP-2, suggesting an active dimer mechanism that is common for these homologous proteins.” Unfortunately, I do not understand precisely what they are suggesting. Is it that the residues involved in HDL binding for SR-B1 and oxidized LDL binding for CD-36 are part of the dimer interface and when they are mutated the dimers don't form, or that they are not part of the dimer interface, but that somehow dimerization influences how these residues might be involved in ligand binding. This is just not clear – a well-designed figure would help. Please clarify.

Fig 3A and P7-8: Under the conditions of the SPR experiments (pH7.5 and 5.0 in Fig) are the monomeric and dimeric (oligomeric) forms of LIMP-2 luminal domain in equilibrium? If ‘monomers’ and ‘dimers’ are separated by size exclusion chromatography (SEC) and then each fraction re-run on SEC – do they equilibrate back into mixtures? What fraction is monomer and what fraction oligomer for the samples used for SPR? Has this been established? [Indeed, in Figure S5 legend the authors suggest there may be ‘monomers’ in the ‘dimer’ sample. Also, oligomer binding in SPR will certainly illicit a larger response in the SPR than monomer when there is 1:1 binding, so it is not surprising that oligomers gave a higher response. The mass of an oligomer is greater than that of a monomer. This must be considered. It is not at all clear what this means, although the differences in ‘monomer’ and ‘dimer’ responses suggest that one or the other is in the preponderance in the different samples used. Also of note, if the ‘dimer’ is a complex mixture of dimers, trimers etc., it is possible that the higher order oligomers bind more tightly than the lower order oligomers. Were the authors surprised that the best ‘binding’ seen in SPR was to PS yet no PS was seen associated with the purified protein by MS?

Figure S5A: what is the difference between K_d and K_d (eq)? Please define.

Table S3: What are the SDs or SEMs for the K_d values shown? Is a K_d of $0.812e-6$ (for dimer

POPS) really significantly different than the values shown for monomer binding: 2.76×10^{-6} ? Please indicate what pH was used for those data shown. Also, please include the data (numerical results) for the pH 5.0 studies in this table (again, give error bars).

P8: please explain the following: "their clear deviations from 1:1 kinetics strongly suggested that LIMP-2 had to dimerize before binding". Where exactly does a reader see these clear deviations?

P8: One could argue from the following that the binding was artifactual rather than of biologic significance: "Furthermore, mutagenesis studies suggested that LIMP-2 uses different sites for binding to PS and PC liposomes (Supplementary Fig. S8)."

FigS8: were monomers or dimers of LIMP-2 used? Not clear. Also, why were the K_d values so similar if there is no PC binding? This is somewhat confusing. Also, please show the data for wild-type LIMP-2 in the same figure for comparison.

P8 "We additionally tested LIMP-2 binding to PS liposomes using NS-EM and DLS. LIMP-2 dimer, but not the monomer, strongly induced clustering of PS liposomes (Fig. 3B), indicating that LIMP-2 dimer tethered liposomes." If there were only one binding site of LIMP-2 luminal domain per monomer for PS liposomes, one would not expect monomers to cluster PS liposomes, but one would expect 'dimers' (oligomers) to cluster them. Thus, the power of the NS-EM and DLS experiment is much less than suggested. This must be made clear in the text.

Please indicate the number of independent replicate experiments (and number of replicates per experiment) used to calculate error bars (e.g., in Fig 4).

P9: The following text suggests that LIMP-2 endocytosis depends on the presence of ligand. Is that true? If so, what ligands? Is there no endocytosis without specific ligand binding? Please clarify. "Live imaging revealed that, upon binding to a ligand, surface-expressed LIMP-2 was readily internalized into the cell, forming endocytic vesicles, and transporting cargo across the cell (Fig. 4B; Supplementary Movie S1),"

Fig 4C&D: please show image of KO cells taken under the same conditions as for WT in panel C. Hard to judge from the bars in panel D, but it seems that most of the signal is background in panel C.

Fig 4 M-T: This reviewer finds these data unimpressive (some cells do and some don't take up exosomes, etc.). These panels are substantially less impressive or significant than the data in the rest of the manuscript. The authors should consider leaving these data out of the manuscript. It seems that these are non-definitive data that reduce, not increase, the impact of the manuscript.

Fig5A-D provide evidence that PS liposome are taken up and delivered to lysosomes by WT MEF cells. The authors should do the same experiment (in parallel) with KO MEF cells. If they did, they could determine what fraction of uptake to lysosomes was LIMP-2 dependent. That would be of considerable interest. Based on Fig 4G, this should be the overwhelming majority of the uptake. The rest of Fig5 E-J is relatively trivial and could easily be moved to the supplement. It has already been well established that LIMP-2 moves the GCase to the lysosome where it is active. This is of very little interest – at least as presented. If the authors can show why this is worth including in this manuscript, they should please make that clear.

Fig 6: In the context of this manuscript, the detailed description of the localization of LIMP-2 in the kidney seems out of place. The results are interesting, but do not provide any particular new insights. Indeed, in young mice the loss of LIMP-2 has no effect on the distribution of injected PS liposomes. The fact that the distribution is altered later in life could be a very indirect consequence of the absence of LIMP-2. This comes as no surprise, as the authors write: "Loss of LIMP-2 expression is associated with kidney dysfunction in humans and mice^{16,38}". The same is true of

BSA processing. So these results are of little interest and provide no new mechanistic or physiologic insights. I recommend the authors consider leaving them out of this otherwise important manuscript.

DISCUSSION:

P14: Please provide a good figure to help with the description shown here (this figure could substitute for the current figures 5 and 6 that need not be in the main manuscript): "Our findings are anchored at the atomic level by the observed conformational switch that converts the GCCase-binding apo LIMP-2 monomer to the lipids-bound dimer that tethers PS liposomes. The LIMP-2 monomer may bind PC liposomes at the entrance of the hydrophobic tunnel after the observed conformational changes that favor dimerization. The LIMP-2 dimer tethers PS liposomes most likely via positive surface charges in both its distal and proximal faces."

P15: Is there any evidence for LIMP-2-mediated selective uptake? "Our results predict that LIMP-2 dimer may facilitate selective uptake of cholesterol from exosomes." If not why is this here?

P15: It is striking that LIMP-2 shows a preference for binding PS. This is not the first time receptors, indeed scavenger receptors have been reported to bind preferentially to PS. The 1st molecularly define cell surface (binding) receptors (and first class B scavenger receptors) for PS were reported in 1995: Rigotti A, Acton SL, Krieger M. The class B scavenger receptors SR-BI and CD36 are receptors for anionic phospholipids. J Biol Chem. 1995 Jul 7;270(27):16221-4. Perhaps this should be cited.

The discussion of the kidney in the Discussion is relatively speculative and, as mentioned above, detracts from the more important structural and functional studies reported in this manuscript.

METHODS:

Please explicitly define Expi293F(HEK) cells and HEK 293GnTi- cells. There is not enough information. Also from where were they obtained.

TopFluor liposomes: Was the incorporation into the liposomes of the fluorescent PS equal to that of fluorescent PC? How was that determined? Or was simply a 10% of fluorescent lipid added to the unlabeled lipid for generation of liposomes. Is the efficiency of fluorescence for the PS and PC labeled lipids identical? Presumably yes, but please indicate.

"Solubilized lipids" Has it been established (ref. please) that BSA-solubilized lipids are not in liposome/micelle form? "BSA-solubilized lipids were made up of 10 mM lipid in 12% w/v BSA with 150 mM NaCl and solubilized by vortexing."

Minor points:

Fig S10 has double images of text.

please do not use abbreviations for methods (MALS, SEC, etc.) without first defining these abbreviations.

Reviewer #2 (Remarks to the Author):

The manuscript by Conrad et al presents their determination of a structure of LIMP-2 in a dimeric form, their analysis of the binding of monomeric and dimeric LIMP-2 to ligands and the role of the LIMP-2 in phospholipid uptake. They show a role for the receptor in uptake of PS liposomes, but they also develop a thesis in which they suggest 'switching' between monomeric and dimeric forms alters interaction with binding partners and that this is shared with other LIMP-2 homologues.

This paper contains a large range of data, obtained using different approaches. While it makes a number of interesting observations, it doesn't work as a whole and I am not convinced that the data strongly supports the idea that the dimeric form of LIMP-2 is physiological and that it switches between monomeric states to alter its ligand specificity. It is also quite a confusing paper in terms of layout and content. It needs a lot of re-writing to make it clear, in particular to give the results section more flow.

The structure of LIMP-2 is well determined and appears of high quality. The authors then describe the structure in comparison with previous LIMP-2 structures, but their figures are hard to follow. It would be good to see a clearer figure that outlines these differences.

The authors claim that their dimeric form is physiologically important. However, it is always challenging to be sure that dimeric forms that appear in crystals are physiological and not due to crystallization artifacts. The authors when proceed to separate out the monomeric and dimeric forms of the protein by size exclusion. It is surprising that this works as systems were both monomer and dimer are seen usually suggest an equilibrium between monomeric and dimeric and purifying one form leads to re-equilibration. Perhaps the authors should comment on this?

The authors then proceed to conduct biophysical binding studies on the 'monomeric' and 'dimeric' forms of LIMP-2. I am far from convinced by these studies or their conclusions. For example, dimeric LIMP-2 showed higher binding levels to POPS than monomeric LIMP-2. But the reported affinities are pretty much identical. What does this mean? Also the authors say that the dimeric form does not bind to GCase, but it does, with an affinity of 5.7microM, only 7-fold more weakly than the monomeric form. There is much about this data which is confusing, and I wonder if it is because their 'monomeric' and 'dimeric' samples are re-equilibrating? Whatever is the cause, it does not strongly support their suggestion that the dimeric form binds liposomes and the monomeric binds to GCase and the protein 'switches'. The data which suggests that LIMP-2 forms dimers in cells is also rather weak, just being based on a single cross-linking experiment; a type of experiment with many risks of artifacts.

In summary, this paper contains some interesting work. However, the biophysical data is not conclusive and does not convince me that their monomer-dimer switch model is correct. This needs more work before it forms a conclusive paper.

Other comments:

The crystallographic table is a little short of information. For example, the CC1/2 is not mentioned. Neither are details of the model quality, such as the number of Ramachandran outliers or the rmsd for bond angles and lengths. I think that Nature journals have a standard proforma for crystallographic tables and it would be better if the authors used this.

'LIMP-2 is required for the uptake of PS liposomes.' It isn't – there is 38% uptake in the KO.

Figure S2 'ligand bound LIMP-2 exists as a dimer in solution' is not quite true. It exists in a mixture of multimeric states. The tail in A towards low volumes also suggests additional multimeric states?

Figure S2A. I am not convinced that these mutants are giving a major difference in dimer percentage. Could the authors do something quantitative to address this?

Figure S2D is hard to understand. Exactly what is being shown here? To what degree do the authors argue the mutants are significantly different?

Figure S3A. What is the 'concatemer' sample?

Figure S8. I was a bit confused by this. The D272 residue is within the hydrophobic cavity of LIMP-2 and is close to the binding site for an isolated PC molecule. However, I can't see how this residue would be able to interact with a PC head group when the lipid is still part of a liposome? How then would it affect the binding of the LIMP-2 to liposomes on the chip? Also the LIMP-2 still binds to low micromolar affinity here.

Figure S10 – something strange has happened to this figure, leading to blurring.

Reviewer #3 (Remarks to the Author):

This referee has been asked to review the organic synthesis part of the manuscript. The fluorescent activity-based probe MWM941 has previously been prepared by Overkleeft et al. (Nat. Chem. Biol. 2010, 6, 907). In the present manuscript, the synthesis has been repeated, but without changing the benzyl protecting group to a benzoyl group. As a result, two steps have been omitted from the synthesis which constitutes a slight improvement. The overall yield of MWM941, however, is still the same since the modified route to MWM941 involves a low-yielding hydrogenolysis of the benzyl ethers at the end. In all, the synthetic procedures in the present manuscript are well-written and the prepared compounds are appropriately characterized.

Reviewer #4 (Remarks to the Author):

This manuscript builds on the discovery and structural characterization of a dimeric, lipid-bound, form of the LIMP2 protein to provide evidence that LIMP2 contributes to the internalization of lipids. Such findings may shed light on the mechanisms whereby LIMP2 mutations cause renal and neurological disease. The data in this manuscript spans a wide range of experiments that range from the x-ray crystallography analysis of a LIMP2 dimer, the demonstration of lipid binding to recombinant LIMP2 extracellular/luminal domain and the characterization of lipid uptake in cell and tissues of LIMP2 knockout mice. The text largely provides a logical and reasonable interpretation of the available data. However, there are some pieces of data that warrant either further investigation or better presentation. These are summarized below.

1. Figure 3 presents anecdotal data for the characterization of interactions between LIMP2 and various lipids. The accompanying results section provides an interpretation of these results. Missing though is the extraction, quantification and presentation of the key points from these examples of raw data.
2. Immunofluorescence labeling for LIMP2 in MDCK cells (Figure 4A) suggests the abundant localization of LIMP2 to the plasma membrane. This conclusion would be strengthened by a demonstration of antibody staining specificity. For example, does this signal disappear in KO cells. Given the ease of generating CRISPR KO cell lines, there is little reason not to provide such a control when proposing a novel protein localization based on antibody staining. Related to this point, Panel 4D compares LIMP2 labeling in WT versus KO MEFs and reports (but does not show primary data) that most of the surface labeling for LIMP2 in MEFs is non-specific. 4B provides support for a sizable plasma membrane pool of LIMP2 but this is based on over-expression and may not reflect the localization of the endogenous protein.
3. Figure 4D, H and L panels contain asterisks on graph that are meant to note statistical significance. However, there is no explanation of the specific statistical tests in the accompanying legend.
4. Details of the quantification method should be provided for Figure 4D, H and L.
5. Figure 5 A-D claim to establish subcellular localization of internalized PS vesicles. However, due

to the crowding of the organelle puncta in these images it seems likely that much of the limited colocalization of PS with lysotracker and MDW941 could simply be due to chance. The text in the Results section provides quantitative measures of colocalization but the details for how this analysis was performed are not provided.

6. The main message is not obvious from visual inspection of Figure 6 I-N. It is claimed that there is reduced PS accumulation in interstitial cells in the KO. This is hard to appreciate from the low magnification, triple stained images that are provided in this figure.

Responses to the reviewers' comments

Reviewer #1 (Remarks to the Author):

Comments to the authors:

The key new findings reported in this manuscript are:

- 1) The structure of the dimeric form of LIMP-2 luminal domain – this result may have important implications for other class B scavenger receptors (CD36 and SR-BI). Note: the authors should clearly indicate in the abstract and elsewhere that they do NOT determine the structure of the whole LIMP-2, but only its luminal domain.
- 2) The monomeric and ‘dimeric’ (possibly oligomeric) forms of the LIMP-2 luminal domain exhibit distinct binding properties – monomer preferred for binding GCase whereas ‘dimer’ preferred for phosphatidyl serine (PS). Again, a very intriguing result.
- 3) Possible modes of binding of lipids to hydrophobic tunnel in LIMP-2 (possible, because the figures were not sufficiently clear to convincingly establish the nature of that binding)

Most of the structure figures need to be redrawn to better show the results they claim they have (see details below). For example, it will be necessary to show clearly the electron density for the lipids bound to the LIMP-2 and clearly show and describe how those lipids are related to the hydrophobic tunnel and protein that lines the tunnel. This is a key issue that they did not address adequately. Also, they need to better show the dimer interface and the hydrophobic cleft they say is generated by dimerization. The structural information is one of the two key types of findings and it has to be presented in a form better accessible to the reader.

We have revised the figures extensively. We have moved a figure showing electron density embedded with lipids to the main text (Fig. 1B). We have also added one more figure (Fig. 1C) of the cross section of LIMP-2 dimer to show the ligand in the hydrophobic tunnel. We labeled peptides interacting with lipids (Fig. 2B, 2C and 2E) and added descriptions on page 8 and 9. We also improved figures showing the dimer interface (Fig. 2F) and the hydrophobic cleft (Fig. 2E) that is generated by dimerization. We also made two movies to clarify structure figures.

The second key finding – that LIMP-2 can serve as a receptor for PS – is definitely worth reporting. However, the authors seem unaware that others have previously reported that class B scavenger receptors (CD36, SR-BI) can function as PS receptors. The first report of this that I could find was Rigotti et al. J Biol Chem. 1995 Jul 7;270(27):16221-4. Given the strong emphasis the authors make on this activity of LIMP-2, it seems worthwhile to put it in the context of earlier related (though not identical) work.

We have included the report, Rigotti et al. J Biol Chem. 1995 Jul 7;270(27):16221-4, as a reference (ref. 17) and mentioned the similarity in PS uptake between LIMP-2, SR-BI and CD36 (P4).

The authors take considerable space describing experiments that tell us what we already know – that LIMP-2 moves bound molecules to the lysosome where GCase (one of its ligands) functions. All of those studies (eg., Fig4M-T) could be moved to the supplement, as they do not provide new insights.

These studies have been moved as requested to the supplementary material (Fig. S12).

There are also some problems with interpreting some of the data, e.g., some cells do and some don't take up exosomes, etc.. These panels are substantially less impressive or significant than the data in the rest of the manuscript. The authors should consider leaving these data out of the manuscript. It seems that these are non-definitive data that reduce, not increase, the impact of the manuscript.

We have removed exosome uptake results from the manuscript.

The authors take considerable space in the results and abstract describing LIMP-2's distribution in the kidney and speculating about its function. However, their own data suggest that the kidney defects they describe in older LIMP-2 KO mice are not seen in younger LIMP-2 KO mice, and thus it is not the absence of LIMP-2 per se, but rather secondary consequences of its absence over time that are responsible for some of the defects they report. Interpreting these findings is problematic and they seem to reduce the impact of the work. The authors may want to consider removing the kidney findings from this manuscript and reporting them later when they have a more definitive story to tell.

Lipid uptake in kidney by LIMP-2 has not been studied before. The phenotype of loss of function of lipid transport of LIMP-2 KO mice kidney is a striking feature. We acknowledge that much more follow up work can be done on this topic. However, we decided to present this data to stimulate interest on this function of LIMP-2.

Specific comments:

TITLE: The title is a bit misleading: "Lipids-bound structure reveals lysosomal integral membrane protein-2 as a phospholipid receptor" It was a series of binding studies in vitro and in vivo that established that LIMP-2 is a phospholipid receptor. The structure did NOT reveal it to be a phospholipid receptor, but rather to show together with MS analysis that there were phospholipids associated with their purified luminal domain. The authors may want to consider revising the title.

We appreciate the suggestion, recognizing the limitations of this title. It has been revised to "Lysosomal integral membrane protein-2 as a phospholipid receptor revealed by structural,

biophysical and cellular studies” to more appropriately reflect the multidisciplinary nature of the collective work.

INTRODUCTION:

PLEASE GIVE MORE EXPLANATION FOR THE FOLLOWING: “Deficient lysosomal sorting of GCase due to LIMP-2 mutations is unlikely to be the primary cause of AMRF.” NOT CLEAR TO THE READER.

We rephrased the statement to “However, since GCase mutations that cause deficiency in enzyme activity do not cause classical AMRF symptoms both clinically and biochemically^{2,3,12}, deficient lysosomal sorting of GCase due to LIMP-2 mutations is unlikely to be the primary cause of AMRF”

P3 – please cite primary papers that are relevant. For example ref #14 is not appropriate: “LIMP-2 belongs to the class B scavenger receptor family that also includes CD36 and SR-B1. CD36 and SR-B1 are primarily receptors for fatty-acids and high-density lipoproteins (HDL), respectively, but their known ligands have expanded to include collagen, low-density lipoproteins (LDL), and phospholipids¹⁴.” Reference 14 is not the proper reference for the CD36 and SR-B1 activities.

The citations were updated to reflect the earlier works in which CD36 and SR-B1’s ligands were identified (Aitman, T.J. et al. Nat Genet 21, 76-83 (1999). Martin, C.A. et al. Protein Science 16, 2531-2541 (2007). Varban, M.L. et al. Proceedings of the National Academy of Sciences 95, 4619-4624 (1998). Rigotti, A., Acton, S.L. & Krieger, M. Journal of Biological Chemistry 270, 16221-16224 (1995), ref. 14-17) and the old reference 14 has been removed for this citation.

P4: “we establish that LIMP-2 binds phospholipids” data for other SRBs? Citation?

The citations were updated and include the statement that directly mentions the binding of phospholipids by SR-B1 and CD36, “...that LIMP-2 binds phospholipids like SR-B1 and CD36” (P4 and new citation from Rigotti et al. (1995) listed above).

RESULTS

P5. The authors must explicitly state very early in the manuscript that they do not have the entire structure with transmembrane and cytoplasmic domains.

We state in the introduction and abstract that we determined the structure of LIMP-2 luminal domain.

The authors point out that “important differences from previous LIMP-2 structures^{14,17,18} were evident in a lipid-bound structure of the LIMP-2 luminal domain”. This seems to be a key point and it would be nice if clear figures showing these differences would be presented. Also, structure of CD36 extracellular domain should also be included in these comparisons.

Indeed there are important differences between our lipid-bound LIMP-2 dimer structure and apo LIMP-2 monomer structure. These differences are: the new structure is a symmetric dimer, vs a monomer; lipids were found bound to LIMP-2 in the new structure; and lipid binding induced large conformational changes as described in the manuscript. We included an additional figure (Figure 2A) and improved the existing figures to reflect these changes. We also included comparisons of LIMP-2 dimer and monomer structure with CD36 extracellular domain structure (P9-10 and Fig. S5). We found that while CD36 adopts very similar structure as the apo LIMP-2 monomer, it has large conformational differences compared to the subunit in LIMP-2 dimer.

P5 They should also state in the text the resolution of the structure and a measure of the quality of the data and the fit of the model. THIS IS ESSENTIAL. This should not be hidden in a supplemental table.

We state at the beginning of the Results section that the structure was determined at 3.0 Å resolution. We moved the X-ray data collection and structure refinement table to the main text (P33). We also added more descriptions of the model fitting on P6.

P5. They correctly state that “Two important differences from previous LIMP-2 structures^{14,17,18} were evident” “Second, clear densities for lipids, presumably acquired from the expression host, were present within each monomer at the hydrophobic tunnel (phosphatidylcholine, PC; cholesterol, CHO. Fig. 1B and Supplementary Fig. S1).” However, it would be appropriate to state here that lipids in the tunnel have been observed in the CD36 structure. (Nat Commun. 2016 Sep 26;7:12837.)

In the revised manuscript we also compared the similarity and differences of the bound lipids in CD36 and LIMP-2 dimer, and discuss their influence in protein conformation and dimerization (P9-10, Fig. S5).

P5 Fig 1 and Sup Fig 1 – These figures do NOT clearly present the relationship between lipids and the hydrophobic tunnel. At least one additional figure clearly showing the lipids with their electron densities and showing them in the envelop of the tunnel and their relationships to the overall structure of the extracellular domain are essential. Also, what are the relationships of the side chains and backbone of the tunnel to the different portions of the lipids (e.g., alcohol in cholesterol, head groups of the lipids. How good are the fits of the electron density to the models of the lipids, particularly that of phospholipids and cholesterol? Are there alternative for the molecules described as lipids (e.g. polypropylene glycol)? These must be shown clearly and described precisely. On approach to this is seen in the figures in the CD36 paper (Nat Commun. 2016 Sep 26;7:12837.).

Improving the figures was one of the main requests of the reviewers, thus we focused on making enhancements to the figures throughout the manuscript, addressing the specific questions raised particularly by Reviewer#1, and using them to better reflect the ideas and data we present. In Figure 1 we moved one figure to the main text (Fig. 1B) showing the lipids with their electron densities and added an additional figure (Fig. 1C) of a cross-section depiction showing the locations of the ligands in the tunnel to improve visualization of their relationships to the overall structure. We added a figure (Fig. 2A) to show the overall conformational changes of subunits

LIMP-2 dimer relative to the LIMP-2 monomer. We described the interactions of phospholipid head group and acyl chains with the tunnel, and the interactions of cholesterol with the tunnel (P8). Peptides that interact with these lipids are labelled (Fig. 2B, 2C, and 2E). If there are specific hydrogen bond interactions, then the side chains/main chains involved are shown (for example, residues interaction with the head group of PC, Fig. 2B). For clarity, side chains involving hydrophobic interactions, which are quite extensive, are generally not shown unless needed, eg, Fig. 2E. We mention that there is no hydrogen bond interaction involving the hydroxyl of cholesterol but are extensive hydrophobic interactions. For fit of electron density for lipids (Fig. 1B), we described (P6) that cholesterol has well defined electron densities and PC molecules have well defined electron densities for the head groups and the SN2 acyl chains, and disordered SN1 acyl chains. Electron density for polypropylene glycol can be better described as to fit a fragment of an acyl chain of a phospholipid (Fig. S1B). We have consulted the CD36 structure paper for figure revision.

Figure 1: The labeling of the lipids in the surface models is very confusing – it is very hard to know what is the protein and what is the lipid – a better way of presenting the lipids than the thick block letters (“CHO”, “PC”) must be used so that it is easy for the reader to see where the lipids are located in these views. Are these lipids really accessible at the surfaces and from each orientation? Also, if the authors showed the relative orientations of the views in 1C, 1D and 1E using arrows and rotations showing the degrees, that would help a reader.

Figure 1A – please label the N and C termini of the two proteins. Also, what is the orientation of 1A relative to 1C,D,E? It seems like the same as in 1C, but not explicitly stated.

We improved Figure 1 with clearer labels and ligand locations. We added one panel of cross-section of the protein to show where lipids bind (Fig. 1C). We also specify the relative orientations of each panel in Figure 1. To improve the contrast we change the color of bound ligand in the surface electrostatic potential presentation from magenta to green, which stands out much better in the colors used (blue, red and white). We removed block letters as way of indicating lipids. Yes, indeed these lipids are accessible at surfaces at the orientations shown and that is one of key difference between dimer structure and monomer structure. We used arrows and rotation degrees to relate these figures. We also state the orientations of these figures explicitly. The N and C- termini are labeled. We also included a Supplementary movie to help visualization.

P5. What is the basis for the following statement: “Both faces have positive charge patches next to hydrophobic areas, a combination that favors interaction with phospholipid bilayers.” Again, a clear figure would help.

We added on P6-P7 this statement: “These cationic patches are formed by clusters of basic residues, R82, R153, R275, R348, K390-K391 and R402 on the distal surface, and K55, K92, R95, K181, K227, K234 and R424 on the proximal face (Fig. 1D, 1F).” These residues are labeled on the surface in these figures.

P5 – Exactly where are the hydrophobic clefts? This is not clear from Fig 1. “but dimerization combines these features and forms the additional hydrophobic clefts.” This is one of several cases in which the figures are inadequate in presenting the structures.

The clefts have been labeled in the added Fig 1C, which was added to show the cross-section of the protein.

P6 and Sup Fig 2A What is the basis of calling the area (left panel) indicated as ‘dimer’ as a dimer and not higher order oligomers? Why is the ‘dimer’ such a broad peak? [Indeed panel 2B suggests that there are higher order oligomers!] Why are the elution volumes so different in the right and left panels of Fig S2A? Define “SEC”. Why does so much of the complex after SDS/reduction run as an apparent dimer? Do you see covalent cross-links in the crystal structure? Please explain the abbreviations in Fig S2C. FigS2D – what about the amounts of cholesterol?

The broad exclusion chromatography (SEC) peak labeled as dimer has been divided into two parts: the peak fractions and the long tail. The peak contains mainly dimers judged by native gel and SEC-MALS. The long tail contains high oligomers. These fractions are labelled accordingly. The elution volumes are different because they were run on different volume columns: the left was run on a HiLoad Superdex 200 16/60 for preparation purposes, while the right was run on a Superdex 200 10/300 for analytical purposes, as indicated in the figure legend. It was typical to see LIMP-2 dimer in the gel samples run with SDS, further attest the stability of lipid bound dimer; the SDS sample buffer used requires a reducing agent to be added separately. There is no evidence of covalent crosslinking in the crystal structure, but may exist in the high oligomers that are formed by dimers which did not crystallize. The abbreviations used in Fig S2C were added into the legend. It was noted in the main text that cholesterol is not able to be detected in this mass spectrometry experiment, but it has also been added to the legend for clarity.

FigS2D – quantitation by LC/MS without internal standards is a very iffy matter. The authors present the reduction in lipid mass in the monomer mutants without any comment about the limitations of such measurements. Also, in the legend they mention some samples of monomer had no lipid, but they don’t indicate if this was a function of different preparations of protein samples or what exactly was the difference between different samples. This reviewer needs more before the claim in the legend can be considered justified. The data and statements are unconvincing.

These samples were standardized by using the same quantity of protein in each sample. An additional figure (Fig. S3D) was added showing an example of comparing the lipid contents of dimer and monomer. We wrote in the legend “Same amounts of protein samples were used. For the example shown, the lipid content in the monomer was ~16+/-8% of that in the dimer (judged by the areas of MH=760.58, the most abundant lipid species), and the monomer and dimer samples were from the same batch of protein expressed, carefully separated on the SEC, and then went through the same ion exchange procedures. Notice the reduction of other lipid species in the monomer.” The difference in the contents of lipids is much larger than the uncertainty in protein concentration determinations, and the reduction in lipid content in monomer was shown for 5 different lipids. The same experimental procedures were used for comparing wild type LIMP-2 and dimerization deficient mutants (Fig. S4B).

P5: “The dimeric structure defined by crystallography concurred with results from multi-angle light scattering (MALS, Supplementary Fig. S2C), structure-guided mutagenesis (Supplementary Fig. S2A), and ligand content analysis (Supplementary Fig. S2D).” Please explain how the ligand content analysis results concur with the dimeric structure.

We acknowledge the poor wording included here, and revised the statement on P7 as follows: “Both the monomer and the dimer species are demonstrably stable following additional ion exchange chromatography after initial separation on SEC, and do not inter-convert in solution after prolonged storage (Fig S3B and S3C). The lack of re-equilibration of these species is most likely due to the facts that lipids binding strongly favors dimer, as described below, while the monomer is probably the energetically stable form of the apo protein. Those bound hydrophobic lipids may not diffuse easily into aqueous solution. Indeed, phospholipids were detected in all dimer samples tested even after extensive detergent wash, while they were significantly less abundant in monomer samples (Fig. S3D).”

FigS3A – please define MOCK. Why was there so much LIMP2 there? Mock usually means mock transfected, but perhaps it means no photoactive amino acids? Experimental details should be provided so that a reader understands what has been done.

We apologize for not defining “mock”. In this experiment, we transfected cells with the concatemer of murine LIMP-2, a GFP expression plasmid (mock) and a murine LIMP-2 expression plasmid and used for detection an anti LIMP-2 antibody, recognizing the c-terminus of LIMP-2. Since we expected a faint signal for the endogenous crosslinked dimer of LIMP-2, we increased the amount of loaded protein in these lanes. Therefore, we detected more monomeric form of LIMP-2 in the mock-transfected cells (all cells were treated with photoactive amino acids). We have updated the figure legend to more accurately depict the experiment completed.

P6: “Modeling studies indicated that the LIMP-2 luminal domain dimer is able to form and to interact with the cell surface when its transmembrane segments are anchored in the cell membrane (Supplementary Fig. S3B).” What is the significance of the dimer model being able “to interact with the cell surface”?

It was rephrased as “Modeling studies indicated that the LIMP-2 luminal domain dimer is able to form on the cell surface. It may simultaneously interact with the cell surface with its proximal face and extracellular lipid vesicles with its distal face when its transmembrane segments are anchored in the cell membrane (Fig. S10F). This potential bridging of cell membrane and lipid vesicles by the LIMP-2 luminal domain may have biological implications, such as facilitating lipids exchange or fusion”. This model was included to prove there is enough space in the terminal residues that are not included in the construct to enable the flexibility required within the exchange from an upright monomer (as LIMP-2 has been shown in previous publications) to the dimeric form identified in the crystal structure. The significance of the capacity to form the dimer while still anchored in the membrane is 1) the crystallized luminal dimer can be part of the full length LIMP-2 dimer in cell, 2) the proximity of the binding sites in the luminal domain of LIMP-2 to the cell membrane itself, which implies potential biological function such as lipids

exchange and/or membrane repair. The bridging of two lipid bilayers by LIMP-2 luminal domain was observed in negative stain EM studies (Fig. 3B). A reference (ref. 18, Engelmann, B. & Wiedmann, M.K.H. Biochimica et Biophysica Acta (BBA) - Molecular and Cell Biology of Lipids 1801, 609-616 (2010)) has been added in the introduction to show that the selective uptake (endocytosis-independent) of lipids is an important and well known function of scavenger receptor B family proteins. Selective uptake of lipids requires approximate of lipid layers with hydrophobic tunnels connecting them.

P6: “The PC binding site is at the entrance half to the hydrophobic tunnel (Fig.1A), with the head group pointed inward (Fig. 2A).” Neither Fig1A or 2A clearly shows the relationship of the PC to the hydrophobic tunnel – please provide a more helpful image. Also regarding “Near the PC head group, D272 and helix α -9 that connects strands β 10 and β 11 are flipped out of the channel to accommodate and anchor the quaternary amine of PC”, it would be helpful to have a figure that shows this more clearly, including appropriate labeling.

Figure 1 & 2 were updated to show these observations more clearly. We hope the added Fig. 1C, Fig.2A and revised Fig. 2B make the relationship of the PC to the hydrophobic tunnel clearer. We also labeled all peptides interacting with PC.

P6 and Fig 1A. The figure does NOT adequately give the reader a feel for how cholesterol is bound: “The CHO molecule is located in the lower half of the hydrophobic tunnel (Fig. 1A).” An image showing how the cholesterol fits the electron density must be shown as well as its relationship to the tunnel and the residues lining the tunnel. The issue of how cholesterol might fit into SR-B tunnels came up in the initial LIMP-2 structure paper and these authors appear to be in a position to provide a definitive answer. Unfortunately, the figures shown do not provide an adequate demonstration of the quality of the fit of the cholesterol model to the electron density nor its relationship to the tunnel and the LIMP-2 residues surrounding it. Again, the authors claim that somehow Fig 2B shows the following: “To accommodate CHO, the loop connecting β 9 and β 10 flips away and is entirely displaced by the α -helical bundle of the opposite subunit (Fig. 2B). Additional peptides are disordered/displaced at or near the CHO binding site, including α 1, β 4 – β 5, β 8 – β 9, and α 10 through α 11 (Fig. 2B). The displacements/disorder of α 1, α 10 and α 11 which cover the CHO site in the monomer structures makes the CHO binding site exposed to both the distal and the proximal faces (Fig. 1C and 1D).” However the loop etc are not even labeled and how can one know what changes occurred when there is no clear comparison of the bound and unbound forms. The exposure of the CHO (not a normal abbreviation for cholesterol, but is for carbohydrate!) and/or its binding site is not at all clear This is yet again an issue where, one presumes, the authors have strong evidence supporting their claim, but they do not show the data (structures) in a way that provides firm support for the claim. And again, on P7 the authors refer to a figure that does NOT clearly illustrate what they say: “At the LIMP-2 dimer interface, the SN2 acyl chains of PC molecules and the glycans on N325 are involved in LIMP-2 interactions (Fig. 2D).” This reviewer URGES the authors to reconsider how they are representing the structure so that a reader will be able to understand (and be convinced by) how the authors draw their conclusions. Otherwise, given the figures provided it can only be taken on faith that much of what they say is supported by their structure. [Note: FigS11B&C ARE good representations that allow the reader to understand what

the authors are saying.]

Both Figures 1 and 2 were extensively updated to be more intuitive and clear for the reader in terms of emphasis on ligand binding, dimerization, and differences from existing structures. We added a new figure (Fig. 2A) to show an overall comparison of subunit of LIMP-2 dimer with the LIMP-2 monomer, with peptides affected by lipids binding and dimerization clearly labeled. We moved a figure (Fig. 1B) to the main text and revised it to show the quality of electron density for cholesterol and PC. We also described in the main text that the fitting for CLR is good. The newly added Fig. 1C shows CLR position in the hydrophobic tunnel. In the revised Fig. 2C peptides interacting with CLR are labelled clearly, and in the newly added Fig. 2D the openings of the CLR binding site on both the distal and proximal faces are shown highlighted. The peptide 9-10, $\alpha 1$, $\beta 4$, $\beta 5$, $\beta 8$, and $\beta 9$ and $\alpha 10$ and $\alpha 11$ are labelled. The abbreviation CHO was changed to CLR, which is used in PDB entries for cholesterol. We revised Fig. 2F to better display the dimer interface, with peptides at the interface, glycan and PC SN2 chain better displayed and labeled.

P6: “In SR-BI, mutagenesis and covalent inhibitor labeling indicated that the equivalent CHO site is critical for the selective uptake of CHO14,25,26.” Please indicate which residues that are in LIMP-2 are equivalent to those in SR-BI are under consideration. Also, while the three citations given are relevant, the correct citations for the mutagenesis of SR-BI by Yu et al are: Biochemistry. 2012 Dec 18;51(50):10044-55; and Proc Natl Acad Sci U S A. 2011 Jul 26;108(30):12243-8.

These residues have been indicated as: “(residues G420 and C384 in SR-B1, corresponding to V415 and A379 in LIMP-2) indicated that the equivalent CLR site is critical for the selective uptake of CLR” (on P8). The citations have been updated to include those specified by the reviewer (references 35,36).

P6: What was the rationale for placing a PPG fragment instead of an acyl chain fragment (or even a water) in the density described: “A fragment of polypropylene glycol precipitant (PPG) was placed in one of the clefts of the LIMP-2 dimer (Supplementary Fig. S1).” Please make the logic clear.

We were just a little bit conservative in claiming ligand identities. We used the word “placed” instead of “modeled” a PPG. The elongated electron density located at the highly hydrophobic cleft can’t be a water molecule. The PPG fragment has been replaced by an acyl chain fragment of PC, based on its shape and the environment it is in, in the updated PDB submission.

P7: The authors appear to make what may be a very important observation about the role of dimerization of SR-B protein in ligand binding, with special reference to CD36 and SR-BI: “This site along $\alpha 5$ is known to be involved in HDL binding for SR-B1 and oxidized LDL binding for CD-3614, but it is completely formed only in the dimer form of LIMP-2, suggesting an active dimer mechanism that is common for these homologous proteins.” Unfortunately, I do not understand precisely what they are suggesting. Is it that the residues involved in HDL binding for SR-B1 and oxidized LDL binding for CD-36 are part of the dimer interface and when they are mutated the dimers don’t form, or that they are not part of the dimer interface, but that somehow

dimerization influences how these residues might be involved in ligand binding. This is just not clear – a well-designed figure would help. Please clarify.

This site was shown to be critical for the fatty acid uptake by CD36, and to the bindings of HDL and LDL in SR-B1 and CD-36, respectively, “and both proteins are known to facilitate selective uptake of phospholipids (Engelmann, B. & Wiedmann, M.K.H. Mol Cell Bio. Lipids 1801, 609-616 (2010))¹⁸. In the monomer, this site is exposed on the surface of these proteins; in the dimer, the hydrophobic clefts are formed and can serve as channels that connect two faces of LIMP-2 to facilitate uptake of lipids”. Due to the highly hydrophobic nature, lipids movements are considered to happen much more readily in a hydrophobic channel than in an aqueous environment. The revised Fig. 1C and 2E should make the argument clear.

Fig 3A and P7-8: Under the conditions of the SPR experiments (pH7.5 and 5.0 in Fig) are the monomeric and dimeric (oligomeric) forms of LIMP-2 luminal domain in equilibrium? If ‘monomers’ and ‘dimers’ are separated by size exclusion chromatography (SEC) and then each fraction re-run on SEC – do they equilibrate back into mixtures? What fraction is monomer and what fraction oligomer for the samples used for SPR? Has this been established? [Indeed, in Figure S5 legend the authors suggest there may be ‘monomers’ in the ‘dimer’ sample. Also, oligomer binding in SPR will certainly illicit a larger response in the SPR than monomer when there is 1:1 binding, so it is not surprising that oligomers gave a higher response. The mass of an oligomer is greater that that of a monomer. This must be considered. It is not at all clear what this means, although the differences in ‘monomer’ and ‘dimer’ responses suggest that one or the other is in the preponderance in the different samples used. Also of note, if the ‘dimer’ is a complex mixture of dimers, trimers etc., it is possible that the higher order oligomers bind more tightly than the lower order oligomers. Were the authors surprised that the best ‘binding’ seen in SPR was to PS yet no PS was seen associated with the purified protein by MS?

Figure S5A: what is the difference between K_d and $K_d(eq)$? Please define.

We have addressed in response above the re-equilibrium question in aqueous solution. LIMP-2 monomer and dimer don’t re-equilibrate in aqueous solution. From the response-concentration SPR curves, the monomer may dimerize on the PC liposomes before binding, but the dimer binds PS liposomes readily. There are small amounts of monomer in the dimer samples used for SPR, and some dimer in the monomer samples (Estimated ~16+/-8% based on native gel and lipid content analysis). The small amounts of monomer in the dimer samples should not impact our conclusions about dimer since it binds PS much stronger. For example, a 10% monomer in the dimer sample with ~5-10% of maximum dimer responses will only count 0.5-1% binding responses of the dimer. The monomer binding to PC should not have interference from the dimer since the latter does not bind to PC. The mass of the dimer sample used in SPR was determined by SEC-MALS to be twice of that of the monomer sample and the dimer had ~7-20 fold larger binding responses than the monomer depending on protein concentrations. We excluded the fractions of the long SEC tail, which contained higher oligomers, in our dimer sample used in SPR. K_d is the kinetic dissociation constant, $K_d=k_d/k_a$, and $K_D(eq)$ is the dissociation constant at equilibrium. Since the system did not reach equilibrium, we dropped the use of $K_D(eq)$. Interaction of protein with liposomes is different from that of binding of individual lipids in the protein. For PS binding, we stated “Surface cationic patches may interact with PS-liposomes,

but may not be well suited for binding individual lipids, which may provide an explanation why PS is not detected in our samples” (P12). Alternatively, PS may be just not available to the secreted LIMP-2 luminal domain in the express system (P7).

Table S3: What are the SDs or SEMs for the Kd values shown? Is a Kd of 0.812e-6 (for dimer POPS) really significantly different than the values shown for monomer binding: 2.76e-6? Please indicate what pH was used for those data shown. Also, please include the data (numerical results) for the pH 5.0 studies in this table (again, give error bars).

The pH of 7.5 was indicated in the methods section and in the Figure legend, but was also added to the Table S2 legend. We added the SDs for the Kd shown in the table. As we stated, the monomer binding to PC liposomes does not follow a 1:1 model, so the weak monomer Kd determination is not as meaningful as we hoped. What is more important is the comparison of their binding responses and the analysis of their kinetic models. At pH 5, LIMP-2 had strong binding responses toward blank L1 chip supporting matrix, and negative binding responses of LIMP-2 were observed relative to the matrix when POPC was immobilized. However, both the monomer and dimer have strong positive binding responses, with the dimer much stronger to PC/PS and PS liposomes. Because of that, it is not meaningful to fit the curves to generate KDs. We stated that SPR responses are in agreements with our SEC-MALS results that LIMP-2 tends to dimerize at acidic pH.

P8: please explain the following: “their clear deviations from 1:1 kinetics strongly suggested that LIMP-2 had to dimerize before binding”. Where exactly does a reader see these clear deviations?

We rephrase the entire SPR section (P10-11). We stated on P10-11 that “Analysis based on the different binding responses and the contents of the residual dimer in the monomer (Fig. S3D) indicated that the monomer binding to PS liposomes was mainly due to the residual dimer in the samples used. Furthermore, the binding of the monomer to PC liposomes, already weak at high protein concentrations and high liposome levels, was undetectable when protein concentrations and liposome levels decreased (Fig.S7), indicating the requirements of dimerization of the LIMP-2 monomer on the liposomes for binding.”. Fig.S7 has been added for this purpose.

P8: One could argue from the following that the binding was artifactual rather than of biologic significance: “Furthermore, mutagenesis studies suggested that LIMP-2 uses different sites for binding to PS and PC liposomes (Supplementary Fig. S8).”

Notice that binding of individual lipid molecule is different from binding the liposome. We removed Fig. S8 and its associated statement, anyway, since the mutant does not address liposome binding behavior.

FigS8: were monomers or dimers of LIMP-2 used? Not clear. Also, why were the Kd values so similar if there is no PC binding? This is somewhat confusing. Also, please show the data for wild-type LIMP-2 in the same figure for comparison.

The binding responses of the mutant for PC are much smaller than for PS. Again LIMP-2 mutant monomer may have to dimerize on liposomes to bind so only small amount of protein can bind yet with similar apparent Kd. We dropped this data set to avoid confusion.

P8 “We additionally tested LIMP-2 binding to PS liposomes using NS-EM and DLS. LIMP-2 dimer, but not the monomer, strongly induced clustering of PS liposomes (Fig. 3B), indicating that LIMP-2 dimer tethered liposomes.” If there were only one binding site of LIMP-2 luminal domain per monomer for PS liposomes, one would not expect monomers to cluster PS liposomes, but one would expect ‘dimers’ (oligomers) to cluster them. Thus, the power of the NS-EM and DLS experiment is much less than suggested. This must be made clear in the text.

We respectfully disagree. We predicted that LIMP-2 dimer should be able to cluster PS liposomes with its two faces before the experiments, and it did (Fig 3B-3C). As we addressed the related question on the significance of interacting with cell membrane (Fig. S4B), LIMP-2 luminal domain dimer tethering liposomes may have significant biological implications that the readers may appreciate, much more than just binding to liposomes. Moreover, as we indicated in our structural analysis, the LIMP-2 monomer has the same two putative PS binding sites in the dimer, plus an additional PC site. The dimer increases PS liposome binding affinity through the avidity effects of dimerization.

Please indicate the number of independent replicate experiments (and number of replicates per experiment) used to calculate error bars (e.g., in Fig 4).

We added “15 measurements of each duplicated experiments” for DLS. We also added “Number of total images observed were 5 and 8 for BSA+PS and LIMP-2 monomer+PS, 16 for POPS, and 12 for PS+LIMP-2 dimer.”

P9: The following text suggests that LIMP-2 endocytosis depends on the presence of ligand. Is that true? If so, what ligands? Is there no endocytosis without specific ligand binding? Please clarify. “Live imaging revealed that, upon binding to a ligand, surface-expressed LIMP-2 was readily internalized into the cell, forming endocytic vesicles, and transporting cargo across the cell (Fig. 4B; Supplementary Movie S1),”

It was rephrased to “...anti-HA antibody bound to surface-expressed LIMP-2-HA was readily internalized...”. Due to technical limitations of the fluorescence microscopy, we can only detect LIMP-2 endocytosis when fluorophore conjugated antibodies were bound. This result did not rule out that LIMP-2 endocytosis might happen without ligand binding. Regardless, our results showcase the dynamic nature of LIMP-2 subcellular distribution and the endocytic trafficking capability of surfaced expressed LIMP-2.

Fig 4C&D: please show image of KO cells taken under the same conditions as for WT in panel C. Hard to judge from the bars in panel D, but it seems that most of the signal is background in panel C.

We added the images of KO cells(Fig. 4G) taken under the same condition as for WT and the images of WT cells treated with IgG isotype control (Fig. 4F) to show the background signal for this set of experiments.

Fig 4 M-T: This reviewer finds these data unimpressive (some cells do and some don't take up exosomes, etc.). These panels are substantially less impressive or significant than the data in the rest of the manuscript. The authors should consider leaving these data out of the manuscript. It seems that these are non-definitive data that reduce, not increase, the impact of the manuscript.

We agree with the reviewer's comment. This set of data was removed from the manuscript.

Fig5A-D provide evidence that PS liposome are taken up and delivered to lysosomes by WT MEF cells. The authors should do the same experiment (in parallel) with KO MEF cells. If they did, they could determine what fraction of uptake to lysosomes was LIMP-2 dependent. That would be of considerable interest. Based on Fig 4G, this should be the overwhelming majority of the uptake. The rest of Fig5 E-J is relatively trivial and could easily be moved to the supplement. It has already been well established that LIMP-2 moves the GCase to the lysosome where it is active. This is of very little interest – at least as presented. If the authors can show why this is worth including in this manuscript, they should please make that clear.

Fig5 A-D was combined with Fig 4 in the revised manuscript. Fig5 E-J was moved to supplementary Fig. S11. We have used several methods to try to determine the lysosomal colocalization fraction in KO MEF. However, for unknown reasons, lysotracker did not form prominent puncta in KO MEF and was not suitable for colocalization analysis. We also tested other lysosomal labeling approaches, such as virus-mediated CellLight lysosome-RFP labeling, dextran feeding, etc. but the labeling intensity was not strong enough in KO MEF for colocalization analysis either. Immunocytochemistry of lysosome markers did show good lysosome distribution in KO MEF but the immuno-staining procedure was not compatible with lipid uptake. Internalized TopFluor-PS was washed away during the staining process. We agree with the reviewer that it is important to determine the LIMP-2 dependent lysosomal transport fraction but due to technical limitations, we cannot resolve this issue.

Fig 6: In the context of this manuscript, the detailed description of the localization of LIMP-2 in the kidney seems out of place. The results are interesting, but do not provide any particular new insights. Indeed, in young mice the loss of LIMP-2 has no effect on the distribution of injected PS liposomes. The fact that the distribution is altered later in life could be a very indirect consequence of the absence of LIMP-2. This comes as no surprise, as the authors write: "Loss of LIMP-2 expression is associated with kidney dysfunction in humans and mice^{16,38}". The same is true of BSA processing. So these results are of little interest and provide no new mechanistic or physiologic insights. I recommend the authors consider leaving them out of this otherwise important manuscript.

Understanding LIMP-2 localization in the kidney provides the insight of which cell type was directly affected by LIMP-2 loss-of-function. In both human AMRF patients and LIMP-2 KO mice, the onset of renal dysfunction in adult life indicates that it is a long-term, direct, and/or indirect consequence of compromised LIMP-2 function. Providing that nothing is known about

the mechanistic dysregulation in AMRF, it would be informative in this manuscript to present the possible correlation of renal failure and a newly established LIMP-2 function in mediating phospholipid transport.

DISCUSSION:

P14: Please provide a good figure to help with the description shown here (this figure could substitute for the current figures 5 and 6 that need not be in the main manuscript): “Our findings are anchored at the atomic level by the observed conformational switch that converts the GCCase-binding apo LIMP-2 monomer to the lipids-bound dimer that tethers PS liposomes. The LIMP-2 monomer may bind PC liposomes at the entrance of the hydrophobic tunnel after the observed conformational changes that favor dimerization. The LIMP-2 dimer tethers PS liposomes most likely via positive surface charges in both its distal and proximal faces.”

The revised figure S10F meets the requirement. In Fig. 10F, we display a monomer with the entrance of the hydrophobic tunnel indicated by a PC modeled there. The cationic patches of both the monomer and dimer are highlighted. Lipid bilayers of the cell membrane and lipid vesicles are modeled to interact with these patches.

P15: Is there any evidence for LIMP-2-mediated selective uptake? “Our results predict that LIMP-2 dimer may facilitate selective uptake of cholesterol from exosomes.” If not why is this here?

This hypothesis is acknowledged to lack sufficient evidence and was removed.

P15: It is striking that LIMP-2 shows a preference for binding PS. This is not the first time receptors, indeed scavenger receptors have been reported to bind preferentially to PS. The 1st molecularly define cell surface (binding) receptors (and first class B scavenger receptors) for PS were reported in 1995: Rigotti A, Acton SL, Krieger M. The class B scavenger receptors SR-BI and CD36 are receptors for anionic phospholipids. J Biol Chem. 1995 Jul 7;270(27):16221-4. Perhaps this should be cited.

It was mentioned in the introduction, but the original 1995 reference was added (ref. 17).

The discussion of the kidney in the Discussion is relatively speculative and, as mentioned above, detracts from the more important structural and functional studies reported in this manuscript.

The Discussion section has been completely rewritten to present a summation of the results and their implications more clearly. We did so in a way to reflect the concerns of Reviewer#1.

METHODS:

Please explicitly define Expi293F(HEK) cells and HEK 293GnTi- cells. There is not enough information. Also from where were they obtained.

Both are explicitly defined. This additional information was included: “Expi293FTM (HEK) cells (ThermoFisher Scientific) and HEK 293GnTi⁻ cells (ATCC® CRL-3022)”

TopFluor liposomes: Was the incorporation into the liposomes of the fluorescent PS equal to that of fluorescent PC? How was that determined? Or was simply a 10% of fluorescent lipid added to the unlabeled lipid for generation of liposomes. Is the efficiency of fluorescence for the PS and PC labeled lipids identical? Presumably yes, but please indicate.

The incorporation of fluorescent PC and PS into liposomes was completed based on measurement by weight of the lipids before extrusion. Any differences in incorporation would not be expected, as the procedure was completed in the same manner and there is no difference in the appended fluorophore. We stated that “TopFluor-conjugated phospholipids which have comparable fluorescence spectra in different phospholipids and are insensitive to pH in the physiological range⁴⁰”.

“Solubilized lipids” Has it been established (ref. please) that BSA-solubilized lipids are not in liposome/micelle form? “BSA-solubilized lipids were made up of 10 mM lipid in 12% w/v BSA with 150 mM NaCl and solubilized by vortexing.”

BSA has more than 6 high affinity binding sites for lipids, thus has been routinely used for lipid solubilization for delivery. Reference (ref. 56) added.

Minor points:

Fig S10 has double images of text.

This is also mentioned by Reviewer #2, and it appears to be upon conversion of the Chemdraw object into pdf. It was resolved by converting the ChemDraw object into a figure first then converting to pdf. It will be attached separately if it is not adequately improved in this final version.

please do not use abbreviations for methods (MALS, SEC, etc.) without first defining these abbreviations.

MALS was defined on page 5, “surface plasmon resonance (SPR), negative stain electron microscopy (NS-EM), and dynamic light scattering (DLS)” on page 4. The SEC acronym definition was added in the legend of Supplementary Figure S2: “Left, LIMP-2 expressed in HEK293-GnTi- cells (incomplete glycosylation) demonstrates a large shift of the size exclusion chromatography (SEC) profiles (Superdex 200, 16-60)”

Reviewer #2 (Remarks to the Author):

The manuscript by Conrad et al presents their determination of a structure of LIMP-2 in a dimeric form, their analysis of the binding of monomeric and dimeric LIMP-2 to ligands and the role of the LIMP-2 in phospholipid uptake. They show a role for the receptor in uptake of PS

liposomes, but they also develop a thesis in which they suggest ‘switching’ between monomeric and dimeric forms alters interaction with binding partners and that this is shared with other LIMP-2 homologues.

This paper contains a large range of data, obtained using different approaches. While it makes a number of interesting observations, it doesn’t work as a whole and I am not convinced that the data strongly supports the idea that the dimeric form of LIMP-2 is physiological and that it switches between monomeric states to alter its ligand specificity. It is also quite a confusing paper in terms of layout and content. It needs a lot of re-writing to make it clear, in particular to give the results section more flow.

We have extensively revised the manuscript to improve its flow and more clearly convey our argument. We have performed additional experiments to explicitly show that when overexpressed, significant amount of full length LIMP-2 exists as dimer in cells, and moreover, endogenous LIMP-2 dimer was detected in three different cell types (Fig. S10). Importantly, the strong cellular lipid uptake preference of LIMP-2 can only be explained by the strong preference of LIMP-2 dimer over monomer for PS liposomes.

The structure of LIMP-2 is well determined and appears of high quality. The authors then describe the structure in comparison with previous LIMP-2 structures, but their figures are hard to follow. It would be good to see a clearer figure that outlines these differences.

We have extensively revised Figures 1 and 2. We also added a figure in the main text (Fig. 2A) and one in the supplementary (Fig. S5A), showing overall comparisons of our structure with the previously determined LIMP-2 monomer and CD36 monomer, respectively.

The authors claim that their dimeric form is physiologically important. However, it is always challenging to be sure that dimeric forms that appear in crystals are physiological and not due to crystallization artifacts. The authors when proceed to separate out the monomeric and dimeric forms of the protein by size exclusion. It is surprising that this works as systems were both monomer and dimer are seen usually suggest an equilibrium between monomeric and dimeric and purifying one form leads to re-equilibration. Perhaps the authors should comment on this?

We are well aware that crystallization could introduce some artifacts; thereby we presented a significant amount of additional data and analysis to support our claim. The dimerization interface covers 900\AA^2 of protein surface of each subunit, which is in the range of a stable dimer. In addition, bound hydrophobic phospholipids induce extensive conformational changes to facilitate dimerization and directly contribute to the dimer interface, consistent with the observed requirements of lipids binding for dimer formation in solution. At the dimerization interface, glycans at N325 are involved in dimer interaction. When expressed in glycosylation deficient mammalian cells, the amount of LIMP-2 decreased significantly (Fig. S3A). When we mutated just two residues at the interface, the dimer content also decreased (Fig. S4A&4B). The dimer structure also explains the binding interactions with liposomes and GCase. We have addressed the same question about re-equilibrium of LIMP-2 monomer and dimer raised by reviewer #1. Those bound lipids are too hydrophobic to easily diffuse out of protein into the aqueous environment and they essentially lock the protein in the dimer form once bound. On the other

hand, the apo protein was always seen as monomer in our hand and in three previous crystal structures, indicating apo LIMP-2 is a stable form.

The authors then proceed to conduct biophysical binding studies on the ‘monomeric’ and ‘dimeric’ forms of LIMP-2. I am far from convinced by these studies or their conclusions. For example, dimeric LIMP-2 showed higher binding levels to POPS than monomeric LIMP-2. But the reported affinities are pretty much identical. What does this mean? Also the authors say that the dimeric form does not bind to GCCase, but it does, with an affinity of 5.7microM, only 7-fold more weakly than the monomeric form. There is much about this data which is confusing, and I wonder if it is because their ‘monomeric’ and ‘dimeric’ samples are re-equilibrating? Whatever is the cause, it does not strongly support their suggestion that the dimeric form binds liposomes and the monomeric binds to GCCase and the protein ‘switches’. The data which suggests that LIMP-2 forms dimers in cells is also rather weak, just being based on a single cross-linking experiment; a type of experiment with many risks of artifacts.

The section of biophysical binding of LIMP-2 monomer and dimer with liposomes and GCCase has been extensively revised. Analysis indicated that there are minor monomer and dimer components in the dimer and monomer samples used in SPR, respectively. Calculations based on their very different binding responses and contents indicate that the monomer binding responses to PS liposomes were mainly due to the minor dimer component. Additional experimental results were added (Fig S6) to show that LIMP-2 monomer had to dimerize on PC liposomes for binding. Determinations of Kd of true monomer for liposomes are not meaningful under these conditions, and the Kds for monomer are dropped to avoid confusions. For GCCase binding, the partial binding responses of the LIMP-2 dimer were from the monomer component, explanation of which was supported conclusively by our SEC-MALS experiments that demonstrated that LIMP-2 dimer does not bind GCCase, but the monomer does. We removed the KD (equi) values for GCCase binding to avoid further confusion.

As we stated, our cross-linking experiments shown were representative of at least three independent experiments. Also cited in our manuscript, full length LIMP-2 dimerization in the cell via its ectodomain has been observed before. In the revised manuscript, we added additional, independent experimental data to show that LIMP-2 forms dimers in cells (Fig. S10B-E).

In summary, this paper contains some interesting work. However, the biophysical data is not conclusive and does not convince me that their monomer-dimer switch model is correct. This needs more work before it forms a conclusive paper.

We hope our revised analysis and statements made it clear to the reviewer that LIMP-2 dimer has much stronger binding affinity and with a strong preference to PS liposomes, while the monomer binds GCCase. We point out that our other experiments such as SEC-MALS (Fig. S9B), negative stain (Fig. 3B) and DLS (Fig. 3C) all support our SPR experiments. We also demonstrated more conclusively that LIMP-2 forms dimer in cells (Fig. S10).

Other comments:

The crystallographic table is a little short of information. For example, the CC1/2 is not mentioned. Neither are details of the model quality, such as the number of Ramachandran outliers or the rmsd for bond angles and lengths. I think that Nature journals have a standard proforma for crystallographic tables and it would be better if the authors used this.

We updated the crystallographic table and moved it to the main text.

‘LIMP-2 is required for the uptake of PS liposomes.’ It isn’t – there is 38% uptake in the KO.

This was edited to: “LIMP-2 contributes significantly PS liposome uptake in cells.”

Figure S2 ‘ligand bound LIMP-2 exists as a dimer in solution’ is not quite true. It exists in a mixture of multimeric states. The tail in A towards low volumes also suggests additional multimeric states?

Yes, the tail labeled dimer does contain higher oligomers, thus the bracketed designation of dimer was changed to show dimer and higher order oligomers separately. The fractions corresponding to the dimer and monomer were purified further from here, and the final products used for SPR analysis are shown by SEC-MALS in Figure S3C.

Figure S2A. I am not convinced that these mutants are giving a major difference in dimer percentage. Could the authors do something quantitative to address this?

We do not argue a major difference in dimer percentage. We refer to them as dimer deficient, as they appear to slightly disrupt the dimer interface. We gave the molecular weights of the main peaks of different species to support claim. We also provided quantitative measurements of lipid contents in the main peaks which we showed to track with dimer contents.

Figure S2D is hard to understand. Exactly what is being shown here? To what degree do the authors argue the mutants are significantly different?

As we demonstrated, lipid contents track with the contents of dimer as measured in our native and SDS gels. The dimer always contained lipids, and the monomer was lipid poor. The amounts of lipids in the dimerization deficient mutants were significantly less than the wild type (~40% reductions).

Figure S3A. What is the ‘concatemer’ sample?

The concatemer is a construct of LIMP-2 with a repeated sequence specifically used for size determination of LIMP-2 oligomers on the gel. The legend for Fig S10A has been updated: “HeLa cells were transfected with a wild-type and a concatemeric LIMP-2 construct, which consists of two repeated full length cDNA sequences of LIMP-2, used to determine the oligomeric state of LIMP-2”. This data supplements the experiment presented in Figure 4, which demonstrated the interaction of two differently tagged LIMP-2 constructs via co-immunoprecipitation using HeLa cells that were co-transfected with LIMP-2-myc and LIMP-2-HA constructs.

Figure S8. I was a bit confused by this. The D272 residue is within the hydrophobic cavity of

LIMP-2 and is close to the binding site for an isolated PC molecule. However, I can't see how this residue would be able to interact with a PC head group when the lipid is still part of a liposome? How then would it affect the binding of the LIMP-2 to liposomes on the chip? Also the LIMP-2 still binds to low micromolar affinity here.

We dropped this data set. We agree with the reviewer that binding of an individual lipid in the hydrophobic tunnel is different from binding to a liposome, and this mutant may not address liposome binding questions. On the other hand, the mutant did bind to PS with far greater binding responses than to PC, consistent with our other results.

Figure S10 – something strange has happened to this figure, leading to blurring.

File conversion issue of this figure was resolved.

Reviewer #3 (Remarks to the Author):

This referee has been asked to review the organic synthesis part of the manuscript. The fluorescent activity-based probe MWM941 has previously been prepared by Overkleeft et al. (Nat. Chem. Biol. 2010, 6, 907). In the present manuscript, the synthesis has been repeated, but without changing the benzyl protecting group to a benzoyl group. As a result, two steps have been omitted from the synthesis which constitutes a slight improvement. The overall yield of MWM941, however, is still the same since the modified route to MWM941 involves a low-yielding hydrogenolysis of the benzyl ethers at the end. In all, the synthetic procedures in the present manuscript are well-written and the prepared compounds are appropriately characterized.

We appreciate the reviewer's comments.

Reviewer #4 (Remarks to the Author):

This manuscript builds on the discovery and structural characterization of a dimeric, lipid-bound, form of the LIMP2 protein to provide evidence that LIMP2 contributes to the internalization of lipids. Such findings may shed light on the mechanisms whereby LIMP2 mutations cause renal and neurological disease. The data in this manuscript spans a wide range of experiments that range from the x-ray crystallography analysis of a LIMP2 dimer, the demonstration of lipid binding to recombinant LIMP2 extracellular/luminal domain and the characterization of lipid uptake in cell and tissues of LIMP2 knockout mice. The text largely provides a logical and reasonable interpretation of the available data. However, there are some pieces of data that warrant either further investigation or better presentation. These are summarized below.

1. Figure 3 presents anecdotal data for the characterization of interactions between LIMP2 and various lipids. The accompanying results section provides an interpretation of these results. Missing though is the extraction, quantification and presentation of the key points from these examples of raw data.

We revised this section related to figure 3 extensively to highlight the key points of our findings. SPR experiments involving lipids interactions are challenging, especially when dealing with proteins that have monomer-dimer conversions. We indicated that the LIMP-2 dimer has exclusive strong binding responses to PS liposomes which follow a 1:1 kinetic model. For the monomer, on the other hand, the binding was either due to residual dimer (PS liposome), or the monomer has to dimerize on PC liposomes for binding. Determinations of binding affinities of true monomer, which are weak, are less meaningful. We hope our analysis was clear this time in getting our arguments through.

2. Immunofluorescence labeling for LIMP2 in MDCK cells (Figure 5A) suggests the abundant localization of LIMP2 to the plasma membrane. This conclusion would be strengthened by a demonstration of antibody staining specificity. For example, does this signal disappear in KO cells. Given the ease of generating CRISPR KO cell lines, there is little reason not to provide such a control when proposing a novel protein localization based on antibody staining. Related to this point, Panel 4D compares LIMP2 labeling in WT versus KO MEFs and reports (but does not show primary data) that most of the surface labeling for LIMP2 in MEFs is non-specific. 4B provides support for a sizable plasma membrane pool of LIMP2 but this is based on over-expression and may not reflect the localization of the endogenous protein.

The specificity of the anti-LIMP-2 antibody was verified by western blotting (Fig. 4D) and immunocytochemistry in LIMP-2 WT and KO MEF (Fig. 4E-G). LIMP-2 expression was detected in the WT cells but not in the KO cells. We included the image of surface labeling in KO MEF (Fig 4G) and the image of IgG isotype control in WT MEF (Fig. 4F) to show the background signal for this set of experiment, and the quantitative comparison of their signals with those of the WT MEF (Fig. 4H)

3. Figure 4D, H and L panels contain asterisks on graph that are meant to note statistical significance. However, there is no explanation of the specific statistical tests in the accompanying legend.

Student's t-test was used for statistical analyses. Figure legend and main text were updated accordingly.

4. Details of the quantification method should be provided for Figure 4D, H and L.

Quantification details were added to the method section- Cell culture and immunocytochemistry.

5. Figure 5 A-D claim to establish subcellular localization of internalized PS vesicles. However, due to the crowding of the organelle puncta in these images it seems likely that much of the limited colocalization of PS with lysotracker and MDW941 could simply be due to chance. The text in the Results section provides quantitative measures of colocalization but the details for how this analysis was performed are not provided.

Although the displayed images shown in 2D projection appeared crowded with organelle puncta, the colocalization analysis was done with 3D confocal z-stack images acquired in the highest resolution. Most puncta appeared to be distinct objects when we used 3D volume rendering

program to visually inspect the localization. It is inevitable that a small fraction of the colocalization may be resulted from unresolved or crowded objects but the colocalization percentage count measured by 3D rendering should reflect the puncta position with marginal error. Quantification details were added to the method section-Phospholipid uptake assay.

6. The main message is not obvious from visual inspection of Figure 6 I-N. It is claimed that there is reduced PS accumulation in interstitial cells in the KO. This is hard to appreciate from the low magnification, triple stained images that are provided in this figure.

High magnification views of regions in Figure 5 I-N were added to the revised manuscript (Fig. 5O-R).

Reviewers' comments:

Reviewer #1 (Remarks to the Author):

General comments:

The authors have done a good job responding to the previous comments. The figures are better (Movie 1 is very helpful), but there are still some problems in the figures that could be addressed to improve the manuscript. Also, this reviewer found the authors' rebuttal regarding the kidney data unconvincing and still recommends that these data be removed from the manuscript as most of these data are uninformative.

The authors may want to look at how the hydrophobic tunnel in the ABCA1 transporter was illustrated in a recent publication (Qian H, Zhao X, Cao P, Lei J, Yan N, Gong X. Structure of the Human Lipid Exporter ABCA1. Cell. 2017 Jun 5. pii: S0092-8674(17)30580-9.) in order to see an example of such a tunnel can be clearly shown to a reader. I suggest this approach be considered by the authors.

There are a number of citations that are not appropriate (e.g., citations of follow-up papers rather than the primary, original papers). Only one of several examples includes citation #20. The first paper suggesting a role of SR-B1 in HCV infection was in 2002 and there were many papers published in 2005-2006 establishing this connection. Citation #20 is from 2007 and not one of the seminal papers, although it is a good one. The authors should review their reference list for such errors and make appropriate corrections. Greater care should be taken by the authors.

Summary:

What does 'preferentially' mean in the following sentence? Compared to what? "Lipid binding alters LIMP-2 from functioning as a glucocerebrosidase-binding monomer toward a dimeric state that preferentially binds phosphatidylserine liposomes" The same concern applies to "preferential" in the following in the Summary: "we demonstrate that LIMP-2 facilitates the preferential uptake of phosphatidylserine into murine fibroblasts" Also, at this point in the abstract the reader does not know what "lipid" in the lipid binding means – this should be clarified.

The following sentence could mislead the readers, because at earlier ages there are no such phenotypes: "At age 6 months in LIMP-2 knockout 32 mice, phosphatidylserine liposome accumulation in interstitial fibroblasts is markedly impaired, 33 and kidney function, gauged by proteinuria and compromised cell membrane integrity, is notably 34 deteriorated." Again, this reviewer feels the kidney data are too preliminary and without adequate explanation to be included in this manuscript.

Introduction:

In the following sentence the authors should consider also citing the tubular proteinemia of the KO mice: "Two lines of evidence suggest that like 69 CD36 and SR-B1, LIMP-2 may also specifically function in lipid trafficking: 1) over-expressing LIMP-2 results in cholesterol accumulation in enlarged hybrid endosome/lysosome compartments²¹; 2) LIMP-2 knockout (KO) mice have a cellular phenotype in the inner ear and ureter reminiscent of an impaired membrane trafficking²²"

The following should indicate that the location of the end of the tunnel is THOUGHT to be near the membrane, but it has not been shown directly:

"A

77 hydrophobic tunnel, reminiscent of the one in cholesteryl ester transfer protein²⁹ traverses the 78 length of the protein and topographically ends near the plasma membrane."

Results:

The following refers to the hydrophobic tunnel, but that is not indicated in Fig 1A -
"two highly distinctive regions of the
102 protein, the α -helical bundle and peptides forming the hydrophobic tunnel (Fig. 1A)."

What do the authors mean by 'at the hydrophobic tunnel': "Second,
103 clear electron densities for one phosphatidylcholine (PC) molecule and one cholesterol (CLR)
104 molecule were present at the hydrophobic tunnel within each monomer (Fig. 1B)."
Do they mean IN the tunnel, or adjacent to the tunnel? Partially buried in the tunnel or completely
buried (not solvent accessible)? This is important, but not clear. A better figure would help here.

Fig 1: Is the PC in a cleft or in the tunnel or both? The text does mention this, but the labeling in
the figure does not differentiate between the cleft bound PRESUMPTIVE PC and the tunnel bound
PC. Also, it is unclear how the disordered SN1 acyl chains are modeled with respect to the tunnel -
again, a figure that clearly shows this is essential. A major feature of this manuscript is the binding
of lipids - thus the reader should be able to get a clear picture of just what is known about the
lipid binding (orientation with respect to the tunnel, where the model is for disordered (not
observed) chains, etc.). This revised version is MUCH better than the original figure, but could be
significantly improved.

Fig 1A: there seems to be a break in the polypeptide chain in the model but there is no comment
about it. What about drawing a dashed line to indicate connectivity that is not shown in the density
map? It is difficult to see where the C and N- termini are and how they might be related to the
plasma membrane (other figure later does show this, but this should also be in Fig 1). Presumably
the 2-fold axis is perpendicular to the plane of the figure, but this is not indicated.

Fig 1B: The reviewer appreciates the authors showing the electron densities for the lipids, but It is
virtually impossible to see how well the PC and cholesterol fit their densities. Please show the
electron densities and the fit models in a larger, higher resolution, image (perhaps in the
supplement) in stereo so that the reader can evaluate the quality of the fit. What do the authors
mean by distal and proximal - not defined in the figure legend? Do they mean facing the
membrane vs facing away from the membrane? Only when you get to a later panel do you learn
about this - how about indicating the membrane as in panel E?

Fig 1C: It is VERY difficult to see how the stick models of the lipids fit into the cross section.
Perhaps better colors or use of black for the stick figures and stronger contrast might help here, or
perhaps a different section or else stereo view.

Fig 1D-F: It is difficult to know which green spheres are cholesterol and which are PC - could there
be some differentiation please? Or some kind of clear labeling?
Fig E to F arrow: seems misleading (in the wrong direction).

Fig S1A - this is a mess - almost impossible to see anything in this - either make it a clear stereo
figure or break it down so that a reader can make sense of this. This is really an unacceptable way
to present the information.

Fig S1B - on what basis is the density claimed to be PC instead of some other type of molecule?
Please explain the rationale.

Movie S1 - This is a BIG help - thank you for doing this. But it looks like the SN1 fatty acyl chains
are just sticking out into solution. Is that true? How can that be? Why are they not modeled into
the tunnel or some other hydrophobic environment? Is there not enough room in the tunnel? If so,
isn't this a problem? Please clarify this issue.

Page 7 and Fig S3:

In the text they write: "132 In agreement with previous work²³, studies of the LIMP-2 luminal domain in solution

133 showed that it exists mainly as a mixture of monomer and dimer on size exclusion
134 chromatography (SEC) (Fig. S3A), native and SDS gels (S3B), and in multi-angle light
135 scattering (MALS) experiments (Fig. S3C)."

The main text should clearly indicate that there are higher order oligomers and shown in Fig S3A/B. Also Fig S3B right seems to show a lot of monomer in the dimer sample. Why is this?

Also, "Both the monomer and the dimer species are

136 demonstrably stable following additional ion exchange chromatography after initial separation
137 on SEC, and do not inter-convert in solution after prolonged storage (Fig S3B and S3C)."

Could the authors show how the fractions from size exclusion chromatography rerun in size exclusion chromatography as stable monomers or dimers?

Fig 2 legend: "873 Residues V415 and A379, the equivalent residue known to be in the cholesterol path in SR-B1,

874 are highlighted" – the cholesterol path in SR-B1 is NOT known. There is a model of the SR-B1 structure based on LIMP-2 and a proposal about the tunnel and the path, but this has not been established – this statement is misleading.

Fig 2D is a very clear illustration of the location of the cholesterol in the tunnel. Thanks to the authors. However, it is not clear how the PC could also fit in here. Could this be shown with a similar panel with both lipids shown? Could this be explained please? Also, is there any evidence for the transport of cholesterol by LIMP-2?

Fig 2F is another example of an image so crowded that it is impossible for the reader to learn anything by looking at it. What is this supposed to show? Please consider redrawing.

Precisely what features of the structure compared to the monomer are we supposed to see in Movie S2 – it was not self-evident to me.

Page 8:

"153 opposite subunit. The SN1 acyl chains of PC exit the hydrophobic tunnel entrance toward the
154 proximal direction and lack additional interactions with LIMP-2." If the SN1 chains were disordered, how can the authors justify this statement?

Page 9: The following is misleading because there is glycosylation in the GnTi cells, it is just truncated. This must be made clear. Do the differences in the WT and mutant oligosaccharide structures help to explain differences in the interface? It is the responsibility of the authors to make this clear:

"187 binding to support dimerization. The SEC profile of LIMP-2 expressed in glycosylation deficient

188 cells (HEK293-GnTi-) shifted significantly toward elution at monomeric molecular weight
189 compared to that of the protein expressed in wild type (WT) cells (HEK293F)(Fig. S3A),"

Fig S5 – the color scheme with pink lipids is Very difficult to see. Please make the lipids black. It would be better if the different panels in S5 were clearly labeled in the Figure indicating which structure is which.

Page 10: Perhaps I am just not thinking clearly, but I don't know how Fig S3C shows that the smaller size of the monomer does not explain the differences seen. Could the authors explain this?:

"215 LIMP-2

216 monomer, on the other hand, bound to different liposomes much more weakly (Fig. 3A), and the

217 weak binding responses cannot be explained by its smaller molecular weight (Fig. S3C)."

Page 11: I think I understand what the authors are saying in the following, but I am not sure that it is convincing. Perhaps the text could provide a more convincing argument?

"Furthermore, the binding of monomer

221 LIMP-2 to PC liposomes, already weak at high protein concentrations and high liposome levels,

222 was undetectable when protein concentrations and liposome levels were decreased (Fig.S6),

223 further indicating a requirement for dimerization of LIMP-2 for liposome binding.

Determination

224 of these weak binding affinities of the LIMP-2 monomer to liposomes was not meaningful under

225 these conditions. In negative control experiments, LIMP-2 did not bind to immobilized LDL (data

226 not shown)."

Page 11: doesn't the following argue for binding of PS to the monomers?:

"232 When liposomes were placed in the mobile phase for SPR binding measurements, they

233 exhibited strong binding to immobilized LIMP-2 at both pH 7.5 and 5.0 after subtraction of 234 background binding. These observations could be attributed to avidity effects resulting from the

235 large sizes of the liposomes, and were effectively irreversible with nearly undetectable rates of

236 dissociation (Fig. S8)." The authors put these data in the supplement, but they seem to contradict their main argument about monomer vs Dimer.

Page 11: As indicated in the previous review, the following could simply be explained by the bivalent binding of dimers acting (like antibodies) to cause aggregation that monomers cannot mediate if monomers have a single binding site and dimers have two of those sites that could bind to different liposomes? This is not a convincing argument:

"237 We additionally tested LIMP-2 binding to PS liposomes using NS-EM and DLS. LIMP-2

238 dimer, but not monomer, strongly induced clustering of PS liposomes (Fig. 3B), indicating that

239 LIMP-2 luminal domain dimer can tether lipid bilayers. This observation was quantified by DLS

240 measurements that demonstrated an increase in particle sizes in PS liposome suspensions

241 when LIMP-2 dimer was added (Fig. 3C)."

Page 13: I could not see the antibody transported across the cell in the figure or the movie:

"When MDCK cells were transfected with a LIMP-2-HA construct and

288 monitored with live imaging, anti-HA antibody bound to surface-expressed LIMP-2-HA was 289 readily internalized into the cell, forming endocytic vesicles and transporting across the cell

(Fig.

290 4B; time lapse, 4C; Movie S3)."

*****Kidney data:

Original comments by reviewer #1:

"The authors take considerable space in the results and abstract describing LIMP-2's distribution in the kidney and speculating about its function. However, their own data suggest that the kidney defects they describe in older LIMP-2 KO mice are not seen in younger LIMP-2 KO mice, and thus it is not the absence of LIMP-2 per se, but rather secondary consequences of its absence over time that are responsible for some of the defects they report. Interpreting these findings is problematic and they seem to reduce the impact of the work. The authors may want to consider removing the kidney findings from this manuscript and reporting them later when they have a more definitive story to tell."

Authors' response in the rebuttal:

"Lipid uptake in kidney by LIMP-2 has not been studied before. The phenotype of loss of function of lipid transport of LIMP-2 KO mice kidney is a striking feature. We acknowledge that much more follow up work can be done on this topic. However, we decided to present this data to stimulate interest on this function of LIMP-2."

This reviewer continues to think that the kidney results (difference at 3 and 6 months) are very complex and confusing and should not be included in this manuscript.

Minor issues:

CLR is NOT a standard abbreviation for cholesterol.

Reviewer #2 (Remarks to the Author):

The manuscript by Conrad et al presents their discovery that LIMP-2 acts as a receptor for PS receptors, presenting a large body of related data to allow them to come to this novel conclusion. There are plenty of interesting findings in this paper. The structural biology is well done and the uptake experiments appear convincing.

The weakness of the paper is the demonstration that LIMP-2 forms dimers in the cell, which is all based on a few pull down experiments rather than a true in vivo experiment. I think that the authors must know this as they have not put any of this data in the main body of the text, but present it all in the SI. There is therefore a bit of a gap between the structural biology and the biophysical work

A lot of the data is still a bit on the scruffy side and the structure figures could do with cleaning up for clarity.

Comments:

P7: The description of the identification of the lipids in the purified CD36 is described later in the text than the description of the structure. This made me ask 'how do they know what to build into the density?' This should be rearranged into a logical order.

P7: Why is the multimerisation data all in SI if this is an important conclusion?

P9: How do you think that the dimerization interface is stabilised by the conformational changes?

P9: The dimerization reduction mutants are not very convincing. They both appear to have a rather minor effect. Is this real?

P10: What is the conclusion from the comparison with CD36? Is it that the same model can't be proposed for this molecule?

Fig. 1 The Figures could do with simplifying to help the reader to see the main point. For example are all of the glycans required on D-F? They make the figure messy.

Fig. 2 Hard to understand with a lot of labels. It makes it hard to clearly see the conformational changes. The movie helps, but from the figures it is not at all clear what the changes are. Perhaps it would be helped by a much 'cleaner' figure such as something like Figure D for both conformational states.

A. what are the blue and green bits? Is it the dimer? In which case why not help the reader by just showing one monomer? Why are there lots of bits of protein missing in the cyan molecule?

F. is very complicated and hard to understand

Fig. 3 The assay shown in B is rather subjective. There might be more clustering in the LIMP-2 dimer sample, but it is hard to be sure as there are also some clusters in the BSA and monomer sample.

Fig 4 A How are you sure that it is on the cell surface? Are these cells permeabilised?

Fig. 5 What is F?

Figure S1A: Why is much of the model not within electron density?

Figure S3C: is the 'monomer pH 5' really a monomer? The mass seems rather intermediate. Also I don't think that changes in shape should affect the molecular weight measurement from MALLS.

Figure S5A: Why show the CIDR domain? It is not needed for the discussion. How about just showing the CD36 and making the figure bigger so that the details can be seen?

Figure S6: it would be nice to see the dimer data next to this monomer data for easy comparison.

Figure S9A: do the authors believe their affinity measurements here? The KD values are not very different (2-fold), but the responses are very different (7-fold). These can't both be true.

Responses to the reviewers' comments

Reviewer #1 (Remarks to the Author):

General comments:

The authors have done a good job responding to the previous comments. The figures are better (Movie 1 is very helpful), but there are still some problems in the figures that could be addressed to improve the manuscript. Also, this reviewer found the authors' rebuttal regarding the kidney data unconvincing and still recommends that these data be removed from the manuscript as most of these data are uninformative.

The authors thank the reviewer for his/her comments. We have improved figure 1 and 2 further to better communicate our results. We have removed the kidney data from the revised manuscript.

The authors may want to look at how the hydrophobic tunnel in the ABCA1 transporter was illustrated in a recent publication (Qian H, Zhao X, Cao P, Lei J, Yan N, Gong X. Structure of the Human Lipid Exporter ABCA1. Cell. 2017 Jun 5. pii: S0092-8674(17)30580-9.) in order to see an example of such a tunnel can be clearly shown to a reader. I suggest this approach be considered by the authors.

We have consulted this recent publication and made an updated figure (Fig. 1B) showing the hydrophobic tunnel and how CLR and PC bind there.

There are a number of citations that are not appropriate (e.g., citations of follow-up papers rather than the primary, original papers). Only one of several examples includes citation #20. The first paper suggesting a role of SR-B1 in HCV infection was in 2002 and there were many papers published in 2005-2006 establishing this connection. Citation #20 is from 2007 and not one of the seminal papers, although it is a good one. The authors should review their reference list for such errors and make appropriate corrections. Greater care should be taken by the authors.

Updated citations are listed below:

#5 Andermann, E. et al.. *Adv Neurol* 43, 87-103 (1986)

#9 Tayebi, N. et al. *Gaucher disease and parkinsonism: a phenotypic and genotypic characterization. Mol Genet Metab* 73, 313-21 (2001).

#10. Aharon-Peretz, J., Rosenbaum, H. & Gershoni-Baruch, R. *Mutations in the glucocerebrosidase gene and Parkinson's disease in Ashkenazi Jews. N Engl J Med* 351, 1972-7 (2004).

#14. Abumrad, N.A., el-Maghrabi, M.R., Amri, E.Z., Lopez, E. & Grimaldi, P.A. *J Biol Chem* 268, 17665-8 (1993).

#15. Acton, S. et al.. *Science* 271, 518-520 (1996).

#16. Tandon, N.N., Kralisz, U. & Jamieson, G.A.. *J Biol Chem* 264, 7576-83 (1989).

#17. Podrez, E.A. et al.. *The Journal of Clinical Investigation* 105, 1095-1108 (2000).

#18. Rigotti, A., Acton, S.L. & Krieger, M.. *J Biol Chem* 270, 16221-4 (1995).

#20 Smith, J. et al.. *Molecular and Biochemical Parasitology* 97, 133-148 (1998).

#21. Scarselli, E. et al. *The EMBO Journal* 21, 5017-5025 (2002).

Summary:

What does ‘preferentially’ mean in the following sentence? Compared to what? “Lipid binding alters LIMP-2 from functioning as a glucocerebrosidase-binding monomer toward a dimeric state that preferentially binds phosphatidylserine liposomes” The same concern applies to “preferential” in the following in the Summary: “we demonstrate that LIMP-2 facilitates the preferential uptake of phosphatidylserine into murine fibroblasts” Also, at this point in the abstract the reader does not know what “lipid” in the lipid binding means – this should be clarified.

These statements have been revised to “Cholesterol and phosphatidylcholine binding alters LIMP-2 from functioning as a glucocerebrosidase-binding monomer toward a dimeric state that preferentially binds anionic phosphatidylserine liposomes over neutral phosphatidylcholine liposomes.” and to “...with a strong preference for phosphatidylserine as substrate over phosphatidylcholine”

The following sentence could mislead the readers, because at earlier ages there are no such phenotypes: “At age 6 months in LIMP-2 knockout mice, phosphatidylserine liposome accumulation in interstitial fibroblasts is markedly impaired, and kidney function, gauged by proteinuria and compromised cell membrane integrity, is notably deteriorated.” Again, this reviewer feels the kidney data are too preliminary and without adequate explanation to be included in this manuscript.

Statements on kidney findings have been removed from Summary.

Introduction:

In the following sentence the authors should consider also citing the tubular proteinemia of the KO mice: “Two lines of evidence suggest that like CD36 and SR-B1, LIMP-2 may also specifically function in lipid trafficking: 1) over-expressing LIMP-2 results in cholesterol accumulation in enlarged hybrid endosome/lysosome compartments; 2) LIMP-2 knockout (KO) mice have a cellular phenotype in the inner ear and ureter reminiscent of an impaired membrane trafficking”

Tubular proteinuria in KO mice was added in the citation (P4)

The following should indicate that the location of the end of the tunnel is THOUGHT to be near the membrane, but it has not been shown directly:

“A hydrophobic tunnel, reminiscent of the one in cholesteryl ester transfer protein traverses the length of the protein and topographically ends near the plasma membrane.”

We inserted the phrase, “is thought to end” in the sentence (P4).

Results:

The following refers to the hydrophobic tunnel, but that is not indicated in Fig 1A - “two highly distinctive regions of the protein, the α -helical bundle and peptides forming the hydrophobic tunnel (Fig. 1A).”

We rephrased the sentence as “First, LIMP-2 in the new structure forms a symmetric dimer (root-mean-square deviation 0.98 Å for all C α atoms in the dimer after applying 2-fold symmetry, Fig. 1A) via contacts in two highly distinctive regions of the protein, the α -helical bundle and the region that forms the hydrophobic tunnel (Fig. 1B)”(P5)

What do the authors mean by ‘at the hydrophobic tunnel’: “Second, clear electron densities for one phosphatidylcholine (PC) molecule and one cholesterol (CLR) molecule were present at the hydrophobic tunnel within each monomer (Fig. 1B).”

Do they mean IN the tunnel, or adjacent to the tunnel? Partially buried in the tunnel or completely buried (not solvent accessible)? This is important, but not clear. A better figure would help here.

We added this sentence “CLR is completely, and PC is partially, buried inside the hydrophobic tunnel (Fig 1B, and see below for detailed descriptions)”. (P5). We also redrew Fig. 1B to better show the bindings of these lipids at the tunnel.

Fig 1: Is the PC in a cleft or in the tunnel or both? The text does mention this, but the labeling in the figure does not differentiate between the cleft bound PRESUMPTIVE PC and the tunnel bound PC. Also, it is unclear how the disordered SN1 acyl chains are modeled with respect to the tunnel – again, a figure that clearly shows this is essential. A major feature of this manuscript is the binding of lipids – thus the reader should be able to get a clear picture of just what is known about the lipid binding (orientation with respect to the tunnel, where the model is for disordered (not observed) chains, etc.). This revised version is MUCH better than the original figure, but could be significantly improved.

We modified Figures 1D-1F and updated the figure legend to differentiate the cleft bound putative and the tunnel bound PC. We rephrased the sentence in P5 to “...the SN1 acyl chains are outside of the hydrophobic tunnel and lack specific interactions with protein, and are thus modelled without definitive electron densities”. We thank the reviewer for this suggestion.

Fig 1A: there seems to be a break in the polypeptide chain in the model but there is no comment about it. What about drawing a dashed line to indicate connectivity that is not shown in the density map? It is difficult to see where the C and N- termini are and how they might be related to the plasma membrane (other figure later does show this, but this should also be in Fig 1). Presumably the 2-fold axis is perpendicular to the plane of the figure, but this is not indicated.

We mentioned disordered/displaced peptides when we discuss lipid binding and dimerization in detail (P8), and the changes in comparison to the apo structure are presented in Figure 2. We added this statement “Several peptides in the LIMP-2 dimer become disordered due to conformational changes caused by lipid binding and dimerization (Fig. 1A).” (P5). We re-drew Fig. 1A with an opposite view so both the N- and the C-termini are closer to the viewer. Drawing dashed lines reduces the clarity of the figure. In the figure legend, we added “The 2-

fold axis of the dimer is perpendicular to the plane of the figure at the center.” (P33).

Fig 1B: The reviewer appreciates the authors showing the electron densities for the lipids, but it is virtually impossible to see how well the PC and cholesterol fit their densities. Please show the electron densities and the fit models in a larger, higher resolution, image (perhaps in the supplement) in stereo so that the reader can evaluate the quality of the fit. What do the authors mean by distal and proximal – not defined in the figure legend? Do they mean facing the membrane vs facing away from the membrane? Only when you get to a later panel do you learn about this – how about indicating the membrane as in panel E?

The initial Fo-Fc maps used for PC and cholesterol modeling were added as Supplementary figures S1B and S1C at high resolution in stereo view. We changed the order of mentioning of distal and proximal as the reviewer suggested.

Fig 1C: It is VERY difficult to see how the stick models of the lipids fit into the cross section. Perhaps better colors or use of black for the stick figures and stronger contrast might help here, or perhaps a different section or else stereo view.

In the revised Fig. 1B we zoomed in the hydrophobic tunnel of one subunit of LIMP-2 dimer to show better how the stick models of the lipids fit into the hydrophobic tunnel.

Fig 1D-F: It is difficult to know which green spheres are cholesterol and which are PC – could there be some differentiation please? Or some kind of clear labeling?

Fig E to F arrow: seems misleading (in the wrong direction).

In the revised Fig 1D-1F, we use yellow color for cholesterol and cyan for PC, and cleft bound PC is indicated. Figure E to F arrow is fixed.

Fig S1A – this is a mess – almost impossible to see anything in this – either make it a clear stereo figure or break it down so that a reader can make sense of this. This is really an unacceptable way to present the information.

The figure was meant to show that large scale conformational changes between LIMP-2 monomer and subunit of LIMP-2 dimer. A stereo figure showing smaller region of model fitting replaces this figure.

Fig S1B – on what basis is the density claimed to be PC instead of some other type of molecule? Please explain the rationale.

We added “Elongated electron densities were also observed at the hydrophobic clefts described below; one allowed modeling of a fragment of a putative PC molecule (Fig. S1D)”. (P5)

Movie S1 – This is a BIG help – thank you for doing this. But it looks like the SN1 fatty acyl chains are just sticking out into solution. Is that true? How can that be? Why are they not modeled into the tunnel or some other hydrophobic environment? Is there not enough room in the tunnel? If so, isn't this a problem? Please clarify this issue.

We stated “...the SN1 acyl chains are outside of the hydrophobic tunnel and lack of specific interactions with protein, and modelled without defined electron densities” (P5). The SN1 chain may interact with the protein surface non-specifically. The SN2 chain is also outside of the tunnel, but “The SN2 chains bend back to the distal direction toward the dimer interface (Fig. 1B) and have hydrophobic interactions with the protein and with each other (Fig. 2B).” (P7). Dimerization and lipid binding are closely associated. There is not enough room in the tunnel to accommodate the SN1 chain. Disordered acyl chains are very common in many other lipid bound protein structures.

Page 7 and Fig S3:

In the text they write: “In agreement with previous work, studies of the LIMP-2 luminal domain in solution showed that it exists mainly as a mixture of monomer and dimer on size exclusion chromatography (SEC) (Fig. S3A), native and SDS gels (S3B), and in multi-angle light scattering (MALS) experiments (Fig. S3C).” The main text should clearly indicate that there are higher order oligomers and shown in Fig S3A/B. Also Fig S3B right seems to show a lot of monomer in the dimer sample. Why is this?

We added the phrase “..some higher order oligomers” in the sentence (P6). Fig S3B right panel is the SDS page gel. Only small amount of dimer survived denaturant treatments.

Also, “Both the monomer and the dimer species are demonstrably stable following additional ion exchange chromatography after initial separation on SEC, and do not inter-convert in solution after prolonged storage (Fig S3B and S3C).” Could the authors show how the fractions from size exclusion chromatography rerun in size exclusion chromatography as stable monomers or dimers?

The SEC-MAL experiments (Fig S3C) were re-runs of SEC on fractions pooled from size exclusion chromatography.

Fig 2 legend: “Residues V415 and A379, the equivalent residue known to be in the cholesterol path in SR-B1, are highlighted” – the cholesterol path in SR-B1 is NOT known. There is a model of the SR-B1 structure based on LIMP-2 and a proposal about the tunnel and the path, but this has not been established – this statement is misleading.

We changed the word “...known..” to “...proposed...” in the revised legend for figure 2.

Fig 2D is a very clear illustration of the location of the cholesterol in the tunnel. Thanks to the authors. However, it is not clear how the PC could also fit in here. Could this be shown with a similar panel with both lipids shown? Could this be explained please? Also, is there any evidence for the transport of cholesterol by LIMP-2?

The revised Fig. 1B and 2D shows clearly how cholesterol and PC occupy different portions of the hydrophobic tunnel. The beginning of the tunnel has specific interactions for the head group of PC, but for cholesterol there are purely hydrophobic interactions. Cholesterol transport by LIMP-2 will be the subject of upcoming projects.

Fig 2F is another example of an image so crowded that it is impossible for the reader to learn anything by looking at it. What is this supposed to show? Please consider redrawing.

We re-drew Fig. 2F with a simplified view showing the contribution of N325 glycan to LIMP-2 dimerization.

Precisely what features of the structure compared to the monomer are we supposed to see in Movie S2 – it was not self-evident to me.

We added the following descriptions in the legends for Movie S2: “The subunits of lipid bound LIMP-2 dimer are shown as green and cyan ribbons, with the apo LIMP-2 monomer (gray ribbons) superimposed to the cyan subunit of the dimer to show conformational differences. Key peptides are labelled.”

Page 8:

“opposite subunit. The SN1 acyl chains of PC exit the hydrophobic tunnel entrance toward the proximal direction and lack additional interactions with LIMP-2.” If the SN1 chains were disordered, how can the authors justify this statement?

We stated “The orientations of the PC molecules at the hydrophobic tunnel are further supported by the specific interactions described below between their head groups and the hydrophobic tunnel” (P5). It is possible to determine the starting positions of SN1 chains from the electron density and interactions with protein, then the density from that point becomes limiting.

Page 9: The following is misleading because there is glycosylation in the GnTi cells, it is just truncated. This must be made clear. Do the differences in the WT and mutant oligosaccharide structures help to explain differences in the interface? It is the responsibility of the authors to make this clear:

“binding to support dimerization. The SEC profile of LIMP-2 expressed in glycosylation deficient cells (HEK293-GnTi-) shifted significantly toward elution at monomeric molecular weight compared to that of the protein expressed in wild type (WT) cells (HEK293F)(Fig. S3A),”

We added the following statement “The expansive mannose-6 glycan, P-Man9GlcNAc2 at N325, that makes intermolecular contacts in the dimer (Fig. 2F) was largely truncated in protein expressed in GnTi- cells (data not shown), which do not produce high-mannose glycans.”(P9)

Fig S5 – the color scheme with pink lipids is Very difficult to see. Please make the lipids black. It would be better if the different panels in S5 were clearly labeled in the Figure indicating which structure is which.

We re-drew Fig. S5 with dark blue instead of pink for FA. We also zoomed in on the CD36/LIMP-2 subunit with bigger figures, and when needed, used spheres instead of sticks to make the lipids more visible. We also labeled different proteins in all panels of this figure.

Page 10: Perhaps I am just not thinking clearly, but I don't know how Fig S3C shows that the smaller size of the monomer does not explain the differences seen. Could the authors explain this?:

“LIMP-2 monomer, on the other hand, bound to different liposomes much more weakly (Fig. 3A), and the weak binding responses cannot be explained by its smaller molecular weight (Fig. S3C).”

We rephrase the statement as “...and the weaker binding responses (<1/10-1/15 of those dimer at the same concentrations at the end of association phases, Fig. 3A) cannot be explained by its smaller molecular weight (1/2.3 of that of the dimer, Fig. S3C).” (P10) to make it clearer. It is known that SPR response is proportional to particle size at any given occupancy.

Page 11: I think I understand what the authors are saying in the following, but I am not sure that it is convincing. Perhaps the text could provide a more convincing argument?

“Furthermore, the binding of monomer LIMP-2 to PC liposomes, already weak at high protein concentrations and high liposome levels, was undetectable when protein concentrations and liposome levels were decreased (Fig.S6), further indicating a requirement for dimerization of LIMP-2 for liposome binding. Determination of these weak binding affinities of the LIMP-2 monomer to liposomes was not meaningful under these conditions. In negative control experiments, LIMP-2 did not bind to immobilized LDL (data not shown).”

We rephrased the statement “The dependence on both high PC liposome immobilization levels and protein concentrations for observations of weak binding responses of LIMP-2 monomer are typical for binding that requires oligomerization”(P10) to make it clearer.

Page 11: doesn't the following argue for binding of PS to the monomers?:

“When liposomes were placed in the mobile phase for SPR binding measurements, they exhibited strong binding to immobilized LIMP-2 at both pH 7.5 and 5.0 after subtraction of background binding. These observations could be attributed to avidity effects resulting from the large sizes of the liposomes, and were effectively irreversible with nearly undetectable rates of dissociation (Fig. S8).” The authors put these data in the supplement, but they seem to contradict their main argument about monomer vs Dimer.

In P10 we added this phrase “as a means to study LIMP-2/lipid interactions in cells” in introducing using SPR for binding. At the end of the paragraph, we added this statement “In a situation that LIMP-2 could pack in the cell membrane as densely as on the SPR chip, which is highly unlikely (see below), both LIMP-2 monomer and dimer would bind PC and PS liposomes through avidity effects of multiple protein molecules binding to liposomes simultaneously.” (P11). We wanted to be thorough in our SPR studies.

Page 11: As indicated in the previous review, the following could simply be explained by the bivalent binding of dimers acting (like antibodies) to cause aggregation that monomers cannot mediate if monomers have a single binding site and dimers have two of those sites that could bind to different liposomes? This is not a convincing argument: “We additionally tested LIMP-2 binding to PS liposomes using NS-EM and DLS. LIMP-2 dimer, but not monomer, strongly induced clustering of PS liposomes (Fig. 3B), indicating that LIMP-2 luminal domain dimer can

tether lipid bilayers. This observation was quantified by DLS measurements that demonstrated an increase in particle sizes in PS liposome suspensions when LIMP-2 dimer was added (Fig. 3C).”

Based on crystal structure analysis (Fig. 1D-F), we proposed that the dimer of LIMP-2 luminal domain tethers lipid bilayers via its two putative lipid interacting sites on the opposite faces (distal and proximal) of the dimer (P12). The two putative lipid interacting faces also exist in the monomer structure, but the dimer has strong affinity for PS due to avidity effects. We add the following statement “Alternative to this structural explanation, we cannot rule out that the monomer may only have one lipid interacting site, but the dimer tethers PS liposomes by combining two sites on the opposite faces of the dimer”. (P12).

Page 13: I could not see the antibody transported across the cell in the figure or the movie: “When MDCK cells were transfected with a LIMP-2-HA construct and monitored with live imaging, anti-HA antibody bound to surface-expressed LIMP-2-HA was readily internalized into the cell, forming endocytic vesicles and transporting across the cell (Fig. 4B; time lapse, 4C; Movie S3).”

The word “across” may be misleading in this case. We rephrase the sentence to “... forming endocytic vesicles and trafficking inside the cell”(P14)

*****Kidney data:

Original comments by reviewer #1:

“The authors take considerable space in the results and abstract describing LIMP-2’s distribution in the kidney and speculating about its function. However, their own data suggest that the kidney defects they describe in older LIMP-2 KO mice are not seen in younger LIMP-2 KO mice, and thus it is not the absence of LIMP-2 per se, but rather secondary consequences of its absence over time that are responsible for some of the defects they report. Interpreting these findings is problematic and they seem to reduce the impact of the work. The authors may want to consider removing the kidney findings from this manuscript and reporting them later when they have a more definitive story to tell.”

Authors’ response in the rebuttal:

“Lipid uptake in kidney by LIMP-2 has not been studied before. The phenotype of loss of function of lipid transport of LIMP-2 KO mice kidney is a striking feature. We acknowledge that much more follow up work can be done on this topic. However, we decided to present this data to stimulate interest on this function of LIMP-2.”

This reviewer continues to think that the kidney results (difference at 3 and 6 months) are very complex and confusing and should not be included in this manuscript.

We removed kidney data from the revised manuscript.

Minor issues:

CLR is NOT a standard abbreviation for cholesterol.

CLR is the 3 letter code used in standard PDB format for cholesterol.

Reviewer #2 (Remarks to the Author):

The manuscript by Conrad et al presents their discovery that LIMP-2 acts as a receptor for PS receptors, presenting a large body of related data to allow them to come to this novel conclusion. There are plenty of interesting findings in this paper. The structural biology is well done and the uptake experiments appear convincing.

The weakness of the paper is the demonstration that LIMP-2 forms dimers in the cell, which is all based on a few pull down experiments rather than a true in vivo experiment. I think that the authors must know this as they have not put any of this data in the main body of the text, but present it all in the SI. There is therefore a bit of a gap between the structural biology and the biophysical work

A lot of the data is still a bit on the scruffy side and the structure figures could do with cleaning up for clarity.

We added additional, orthogonal experimental data demonstrating that LIMP-2 forms dimers in cells (Fig. 4) and improved the structure figures (Fig. 1, 2) for clarity. We put the new data which directly shows the existence of LIMP-2 dimer, localized in lysosomes, using the split-GFP method, in the main text.

Comments:

P7: The description of the identification of the lipids in the purified CD36 is described later in the text than the description of the structure. This made me ask ‘how do they know what to build into the density?’ This should be rearranged into a logical order.

We mentioned in LC/MS experiments that “Cholesterol did not ionize well under the conditions of this analysis, and its presence was neither confirmed nor contradicted” in P6. We also rephrased the statements in P5 “... the buried head groups and the SN2 acyl chains of PC molecules have well defined electron densities... the orientations of the PC molecules at the hydrophobic tunnel are further supported by the specific interactions described below between their head groups and the hydrophobic tunnel.”. These lipids were first identified by X-ray analysis with as much certainties as possible and we then moved on to further identify them using LC/MS and LC/MS/MS.

P7: Why is the multimerisation data all in SI if this is an important conclusion?

In the revised manuscript we put data showing the multimerisation of LIMP-2 in live cells in the main text as figure 4.

P9: How do you think that the dimerization interface is stabilised by the conformational changes?

We added following statements “Conformational changes of $\alpha 9$ also contribute to formation of this protein-protein interface (Fig. 2B) and binding of PC at the hydrophobic tunnel (Fig. 2D). In addition to these protein-protein interactions, the acyl chains of the PC molecules bound at the hydrophobic tunnel (Fig. 2B) and at the clefts (Fig. 2E) also contribute extensively to the stabilization of LIMP-2 dimer, explaining the requirement for lipid binding to support dimerization. Re-arrangements of $\beta 8-11$ are needed for the second subunit to approach the first subunit (Fig. 2C) and to form the PC binding hydrophobic clefts (Fig. 2E). ” (P8, P9).

P9: The dimerization reduction mutants are not very convincing. They both appear to have a rather minor effect. Is this real?

We rephrased the statement “..mutants... demonstrated large decreases (3.0 and 9.5 kDa, respectively) in the molecular weights measured by SEC-MALS in comparison to WT, significant larger than the experiment uncertainties (<0.8 kDa) (Fig. S4A)” in P9. The decrease in lipid content of these two mutants is also significant (>36%, with uncertainty of ~1%, Fig. S4B). We introduced a minimum number of mutations, in a loop, to avoid large protein structure changes. We did state that the mutations only have limited impact on the dimerization. The same mutations were generated in the newly added split-GFP experiments for detection of full length LIMP-2 dimer in live cells, and the decrease of LIMP-2 dimer in cells is also significant ($p < 0.01$, Fig. 4).

P10: What is the conclusion from the comparison with CD36? Is it that the same model can't be proposed for this molecule?

We added this statement “For CD36 and SR-BI to form and function as dimers, bulky lipids such as cholesterol and phospholipids are likely needed to occupy their hydrophobic tunnels to induce similar conformational changes observed in LIMP-2.”(P9-10). As mentioned in the manuscript, CD36 and SR-BI have also been reported to form and seem to have biological functions as dimer.

Fig. 1 The Figures could do with simplifying to help the reader to see the main point. For example are all of the glycans required on D-F? They make the figure messy.

We removed glycans in the revised Fig. 1D-F, and mention it in the legend. We also redrew Fig. 1B to show the hydrophobic tunnel.

Fig. 2 Hard to understand with a lot of labels. It makes it hard to clearly see the conformational changes. The movie helps, but from the figures it is not at all clear what the changes are. Perhaps it would be helped by a much ‘cleaner’ figure such as something like Figure D for both conformational states.

We removed most of the labels in Fig 2, but kept minimal numbers of labels that are essential for discussions. With protein ribbons it is clear how PC and CLR bindings induce conformational changes. We revised Fig 2D to show the hydrophobic tunnels for both conformational states.

A. what are the blue and green bits? Is it the dimer? In which case why not help the reader by just showing one monomer? Why are there lots of bits of protein missing in the cyan molecule?
F. is very complicated and hard to understand

We added in the revised legends for Fig. 2 that cyan ribbons are from one LIMP-2 subunit, and green ribbons are those of $\alpha 4$, $\alpha 5$ and $\alpha 7$ of the second subunit in the dimer (P31). Showing small portion of the second subunit helps readers locate the dimerization interface and understand how conformational changes are required for both lipid binding and dimerization and how these two are associated with each other. We did mention that “Additional peptides are disordered/displaced due to CLR binding, including ... (Fig. 2C).” (P7). We discussed that these disorder/displacements have implicated functional consequences (Fig. 2D) (P7,8), and are required for dimerization (P8,9), not necessarily reflecting the quality of the structure. Fig. 2F has been simplified to show only contributions to dimerization by glycans at N325.

Fig. 3 The assay shown in B is rather subjective. There might be more clustering in the LIMP-2 dimer sample, but it is hard to be sure as there are also some clusters in the BSA and monomer sample.

Particles can stack in the Z-direction and appear to be close in 2D of the negative stain field, but the real clustering in the dimer sample, like the grape bundle shaped clusters and clear touching of two liposomes as indicated in the revised Fig. 3B, do not appear in monomer and BSA samples. We rephrased the legend for Fig. 3B as “Red arrow indicates a clear example of two liposomes forming a tight association with a clear boundary bridged by LIMP-2 dimer (insert for an enlarged view). Blue arrow indicates a grape bundle shaped cluster of multiple liposomes. These features are not observed in any images of BSA and LIMP-2 monomer samples, which were used as negative controls”. We used DLS (Fig. 3C) to quantify the increase in PS liposome sizes due to protein tethering. As indicated in Fig. 3C legend, “the average radii are 50.6 ± 0.4 , 6.53 ± 0.06 , 68.6 ± 1.1 , 3.84 ± 0.02 , 52.4 ± 4 Å for PS liposome, LIMP-2 dimer, PS+LIMP-2 dimer, BSA and PS+BSA, respectively...” (P35). It is known that BSA binds liposomes, but in DLS it clearly does not increase liposome sizes significantly.

Fig 4 A How are you sure that it is on the cell surface? Are these cells permeabilised?

Fig. 4A is a permeabilized staining of MDCK epithelial cells that form a confluent monolayer. Monolayer formation is due to contact inhibition of adjacent cells, in which the surface membrane becomes tightly packed borders. We did verify the surface membrane border position under the bright field phase contrast microscopy before taking the fluorescent confocal images for Fig. 4A.

Fig. 5 What is F?

Fig. 5F is the bright field phase contrast view of the imaging area showed in Fig. 5E. The phase contrast view provides the anatomical position of proximal and distal tubules.

Figure S1A: Why is much of the model not within electron density?

We indicated in the original figure S1A legend that those peptides outside of electron density (gray ribbons) are of the superimposed apo LIMP-2 monomer. They meant to provide experimental evidences for the conformational changes between subunits of LIMP-2 dimer and LIMP-2 monomer. The figure has been re-drawn to a simpler version without showing LIMP-2 monomer to avoid confusions.

Figure S3C: is the ‘monomer pH 5’ really a monomer? The mass seems rather intermediate. Also I don’t think that changes in shape should affect the molecular weight measurement from MALLS.

We rephrase the statement about LIMP-2 samples at pH 5.0 to “At pH 5.0, the molecular weights of LIMP-2 samples that behave normally as monomer and dimer at pH 7.2 increase to 102.0 ($\pm 0.05\%$) and 163 ($\pm 0.03\%$) kDa, respectively, indicating that LIMP-2 has a strong propensity to dimerize at acidic pH” (P8, Supplementary Material) to avoid confusion.

Figure S5A: Why show the CIDR domain? It is not needed for the discussion. How about just showing the CD36 and making the figure bigger so that the details can be seen?

In the revised Fig.S5A we zoom in with bigger figures on CD36 or one subunit of LIMP-2 dimer. Only small portions of their binding partners are shown to orient the readers with the binding interface.

Figure S6: it would be nice to see the dimer data next to this monomer data for easy comparison.

Fig. S6 shows binding of monomer to POPC liposomes at lower protein concentrations and lower lipid immobilization levels. In the revised figure dimer data was added next to the monomer one. As we state here and in the main text, LIMP-2 dimer does not have any binding toward PC liposomes under all conditions tested.

Figure S9A: do the authors believe their affinity measurements here? The K_D values are not very different (2-fold), but the responses are very different (7-fold). These can’t both be true.

We rephrase the statement in the figure legend regarding dimer binding to GCCase to “The responses of LIMP-2 dimer sample are only fractions of those of monomer at the same concentrations. The small responses of the dimer can be best fitted with bindings only from fraction of monomer species ($\sim 1/7$) that exists in the dimer sample, resulting very similar K_D but fraction of R_{max} of the monomer.” and only list the monomer binding parameters which were the truly measured data to avoid confusions (P16, SI). In the figure legend and main text, we clearly stated that the dimer does not bind GCCase, as further supported by experiments in Fig. S9B.

We thank the Reviewers for their comments, which have strengthened and improved the manuscript.